# Anticarcinogenic Potency of EF24: An Overview of Its Pharmacokinetics, Efficacy, Mechanism of Action, and Nanoformulation for Drug Delivery

**DOI:** 10.3390/cancers15225478

**Published:** 2023-11-20

**Authors:** Iliyana Sazdova, Milena Keremidarska-Markova, Daniela Dimitrova, Vadim Mitrokhin, Andre Kamkin, Nikola Hadzi-Petrushev, Jane Bogdanov, Rudolf Schubert, Hristo Gagov, Dimiter Avtanski, Mitko Mladenov

**Affiliations:** 1Department of Animal and Human Physiology, Faculty of Biology, Sofia University ‘St. Kliment Ohridski’, 1504 Sofia, Bulgaria; i.sazdova@biofac.uni-sofia.bg (I.S.); m_keremidarska@uni-sofia.bg (M.K.-M.); hgagov@uni-sofia.bg (H.G.); 2Institute of Biophysics and Biomedical Engineering, Bulgarian Academy of Sciences, 1113 Sofia, Bulgaria; daniadim@yahoo.com; 3Department of Fundamental and Applied Physiology, Russian States Medical University, 117997 Moscow, Russia; mitrokhin_vm@rsmu.ru (V.M.); andrey.kamkin@rsmu.ru (A.K.); 4Institute of Biology, Faculty of Natural Sciences and Mathematics, Ss. Cyril and Methodius University, 1000 Skopje, North Macedonia; nikola@pmf.ukim.mk; 5Institute of Chemistry, Faculty of Natural Sciences and Mathematics, Ss. Cyril and Methodius University, 1000 Skopje, North Macedonia; jane.bogdanov@pmf.ukim.mk; 6Institute of Theoretical Medicine, Faculty of Medicine, University of Augsburg, Universitätsstrasse 2, 86159 Augsburg, Germany; rudolf.schubert@med.uni-augsburg.de; 7Friedman Diabetes Institute, Lenox Hill Hospital, Northwell Health, 110 E 59th Street, New York, NY 10022, USA

**Keywords:** anticancer agent, curcumin, EF24, pharmacokinetics, mechanism of action, nanoformulation

## Abstract

**Simple Summary:**

This study summarizes the current state of knowledge about the anticancer potential of the curcumin analog EF24, including its pharmacokinetic profile, therapeutic efficacy, underlying mechanisms, and advancements in nanoformulations for effective drug delivery. EF24 and its nanoformulations hold great promise as novel and effective therapeutic strategies for cancer treatment and provide hope for patients to fight this devastating disease. More preclinical and clinical investigations are needed to fully unlock their potential and integrate them into clinical practice.

**Abstract:**

EF24, a synthetic monocarbonyl analog of curcumin, shows significant potential as an anticancer agent with both chemopreventive and chemotherapeutic properties. It exhibits rapid absorption, extensive tissue distribution, and efficient metabolism, ensuring optimal bioavailability and sustained exposure of the target tissues. The ability of EF24 to penetrate biological barriers and accumulate at tumor sites makes it advantageous for effective cancer treatment. Studies have demonstrated EF24’s remarkable efficacy against various cancers, including breast, lung, prostate, colon, and pancreatic cancer. The unique mechanism of action of EF24 involves modulation of the nuclear factor-kappa B (NF-κB) and nuclear factor erythroid 2-related factor 2 (Nrf2) signaling pathways, disrupting cancer-promoting inflammation and oxidative stress. EF24 inhibits tumor growth by inducing cell cycle arrest and apoptosis, mainly through inhibiting the NF-κB pathway and by regulating key genes by modulating microRNA (miRNA) expression or the proteasomal pathway. In summary, EF24 is a promising anticancer compound with a unique mechanism of action that makes it effective against various cancers. Its ability to enhance the effects of conventional therapies, coupled with improvements in drug delivery systems, could make it a valuable asset in cancer treatment. However, addressing its solubility and stability challenges will be crucial for its successful clinical application.

## 1. Introduction

### 1.1. Structurally Related Mechanisms of Action

The curcumin (CUR) analog EF24, ((3E,5E)-3,5-bis[(2-fluorophenyl)methylene]-4-piperidinone), is a small-molecule pharmaceutical agent exhibiting promising therapeutic potential across various disorders. Its distinctive structural attributes significantly contribute to its biological efficacy and mechanisms of action. EF24 belongs to the so-called monocarbonyl analogues of curcumin (MACs), where the diketo moiety is replaced by a single keto group that is actually a cross-conjugated dienone that is also in conjugation with two aryl groups [1]. By virtue of this (double) Michael acceptor functionality, EF24 has the capacity to engage with nucleophilic species/residues through a nucleophilic addition process, wherein an electron-rich nucleophile reacts with the electron-deficient enone. This chemical interaction culminates in the formation of a covalent linkage between EF24 and the nucleophilic species, subsequently inducing a modified protein structure [1,2]. It has been pointed out that these MACs tend to react preferentially with thiols rather than with species with amino or hydroxyl groups. Depending on the individual protein involved and the location of modification, the nucleophilic addition reaction with EF24 might have a variety of outcomes. This reaction may elicit a spectrum of consequences, including suppression or activation of protein function, alterations in protein conformation, disruption of protein–protein interactions, or regulation of enzymatic activity. The precise outcome hinges on both the reactivity of the nucleophilic residue and the functional significance of the modified protein within cellular processes [2].

The Michael acceptor functionality of EF24 enables the covalent modification of specific proteins through nucleophilic addition reactions, providing a unique mechanism for modulating cellular processes and potential therapeutic applications in various disease contexts. Also, the unique feature of this system is that the addition of thiols may be reversible, as has been shown by Sun et al. in the case of EF24 and the biologically relevant tripeptide glutathione [3]. The exact mechanism of action, the reversibility of the 1,4-addition, and the involvement of EF24 and other MACs in the disulfide thiol exchange(s) have to be thoroughly investigated.

In comparison to other CUR analogues and some natural products, EF24 offers distinct advantages, primarily stemming from its engineered properties, pro-apoptotic effects, and its potential role as a targeted therapy in the context of cancer treatment [4,5,6,7,8,9]. Actually, it is crucial to acknowledge that most of the mentioned analogues come with certain limitations. These limitations can be attributed to either weak bioavailability or a trade-off between their more pronounced antioxidant or anti-inflammatory effects and their efficacy in combating cancer [4,5,6,7,8,9]. Taking into consideration these inherent characteristics and limitations, EF24 stands out as one of the most suitable and potent options. This assessment is grounded in its unique attributes, enhanced bioavailability, anti-inflammatory effects, and well-documented pro-apoptotic properties, which collectively position it as a highly promising candidate for addressing cancer-related concerns.

### 1.2. Lipophilic Properties

EF24 demonstrates pronounced lipophilicity, indicated by its strong affinity for lipids and its capacity to dissolve in lipid-rich environments or nonpolar solvents [10,11]. These characteristics predominantly stem from EF24’s structure, encompassing hydrophobic segments and the absence of polar functional groups (Figure 1). The lipophilic quality inherent to EF24 significantly facilitates its efficient penetration through cellular membranes, encompassing both the cell membrane’s lipid bilayer and numerous intracellular compartments. The ability to traverse lipid barriers is pivotal for EF24’s bioavailability and therapeutic efficacy [11]. Moreover, EF24’s lipophilic nature profoundly influences its dispersion within the organism. Lipophilic compounds tend to accumulate in lipid-rich tissues or compartments, such as adipose tissue, driven by their strong affinity for lipid-laden environments [12,13]. This aforementioned characteristic has the potential to influence EF24’s pharmacokinetics and pharmacodynamics, encompassing processes like absorption, distribution, metabolism, and excretion [13]. It is important to note that the lipophilic properties of EF24 can also influence its solubility. Being lipophilic, EF24 has limited solubility in aqueous environments, necessitating the use of appropriate solvents or formulation strategies to enhance its solubility and delivery [13,14].

The lipophilic nature of EF24 plays a crucial role in its cellular uptake, distribution, and interactions in the body. It contributes to its ability to cross lipid barriers, target specific intracellular compartments, and modulate cellular processes. Prolonged accumulation in lipid-rich organs can lead to toxicity over time, as it may affect their normal function. Understanding and optimizing the lipophilic properties of EF24 could be essential in its development as a therapeutic agent.

### 1.3. EF24-Mediated ROS Modulation

The EF24 compound has been shown to regulate the production and removal of reactive oxygen species (ROS) [15]. In some circumstances, it serves as an antioxidant, effectively mitigating oxidative stress, while in others, it orchestrates the generation of ROS to facilitate apoptosis or hinder tumor proliferation [15] (Figure 2A). Different studies have demonstrated EF24’s capacity to amplify the production and effectiveness of various intracellular enzymes engaged in ROS scavenging, culminating in enhanced detoxification of ROS [15,16]. This compound also possesses the ability to initiate pivotal cellular pathways governing antioxidant responses, particularly the route mediated by nuclear factor erythroid-related factor 2 (Nrf2) [17]. Activation of Nrf2 triggers enhanced expression of numerous genes related to antioxidant and detoxification processes, including apurinic endonuclease (APE), superoxide dismutase (SOD), and 8-oxo guanine DNA glycosylase (OGG). This orchestrated activation facilitates the effective removal of ROS and the consequent reduction of oxidative stress [17]. Additionally, it has been shown that EF24 has the ability to impede the activity of enzymes implicated in the formation of ROS, notably NADPH oxidase (NOX) [18]. The NOX enzymes are responsible for generating ROS as a by-product, and the inhibition of these enzymes can potentially decrease the total production of ROS. In a study conducted in 2014, Roy et al. [19] documented the protective effects of EF24 on protein disulfide isomerase (PDI), a crucial chaperone found in the endoplasmic reticulum (ER) and pivotal in oxidoreductase activity. Their findings revealed that EF24 exhibited the capacity to safeguard PDI from damage inflicted by ROS. Furthermore, EF24 was found to hinder ROS formation and induce the transcription of genes regulated by the antioxidant response element (ARE) in human ovarian cancer cells (IGROV1 and SK-OV-3) [20]. Moreover, in another study, the concurrent administration of EF24 and SN38 (antineoplastic drug, 7-Ethyl-10-hydroxycamptothecin) led to a significant reduction in superoxide concentration within malignant tissues [21].

Nevertheless, the authors who originally designed and synthesized EF24 noted its ability to induce ROS production in MDA-MB-231 human breast cancer and DU-145 human prostate cancer cells [2]. Similarly, EF24 has been reported to stimulate ROS production in gastric cancer cells [22] and exhibit synergistic antitumor effects when combined with rapamycin [23] or Akt inhibitors [24]. In the case of human colon cancer lines, the same research group reported a similar ROS-inducing effect of EF24 in HCT-116 and SW-620 cells, whereas HT-29 cells showed only a moderate response [25]. While the generation of ROS may initially appear as a potential drawback due to oxidative stress, EF24 utilizes this phenomenon to its advantage by affecting multiple pathways, ultimately promoting anticancer and anti-inflammatory effects. Nevertheless, the specific effects and consequences of this crosstalk may vary depending on the cellular context and the concentrations of EF24 used, and further research is needed to fully understand these interactions. The precise mechanisms underlying EF24-induced ROS scavenging may also vary depending on the specific cellular context and the involved ROS species. Further research is needed to fully elucidate the detailed molecular mechanisms by which EF24 modulates ROS scavenging and its implications for disease pathogenesis and therapeutic interventions.

### 1.4. Anti-Inflammatory Effects

A recent publication reported that EF24 has anti-inflammatory properties, thereby suggesting its potential utility as a therapeutic agent against inflammation [17]. The same study has shown that EF24 effectively hinders the activation of many pro-inflammatory signaling pathways, such as NF-κB, which have a pivotal role in the process of inflammation (Figure 2B). EF24-induced inhibition of NF-κB activation causes a significant reduction in pro-inflammatory cytokines, chemokines, and inflammatory mediators [17,26,27]. Actually, it has been shown that EF24 has the ability to modulate the transcription of genes associated with the inflammatory response [17]. The expression of pro-inflammatory genes, including inducible nitric oxide synthase (iNOS) and cyclooxygenase-2 (COX-2), is downregulated by EF24. In addition, previous studies have demonstrated the ability of EF24 to decrease the infiltration of inflammatory cells into damaged tissues [28]. This mechanism of action might contribute to the alleviation of inflammation by inhibiting the production of inflammatory mediators from these cells [28]. The potential of EF24 to effectively scavenge ROS and regulate oxidative stress may potentiate its consequential impact on inflammation [1]. The relationship between oxidative stress and inflammation is highly interconnected, and the compound EF24 has the potential to mitigate inflammatory responses by decreasing the levels of ROS and minimizing oxidative harm [1]. In addition, it has been shown that EF24 has the ability to impede the synthesis and secretion of pro-inflammatory cytokines, such as interleukin-1β (IL-1β), tumor necrosis factor-alpha (TNF-α), and interleukin-6 (IL-6), which are pivotal in the initiation and augmentation of inflammatory reactions [17,26,27,28].

The specific mechanisms underlying the anti-inflammatory effects of EF24 may vary depending on the cellular context and the specific inflammatory pathways involved. Further research is needed to fully elucidate the detailed molecular mechanisms by which EF24 exerts its anti-inflammatory properties and to determine its potential therapeutic applications in inflammatory diseases.

### 1.5. Anticarcinogenic Activity

EF24 exhibited remarkable anticancer efficacy across numerous preclinical investigations, thereby underscoring its potential as a highly promising therapeutic agent against cancer [1] (Figure 2C). Notably, studies have illuminated EF24’s capability to initiate apoptosis in osteosarcoma cells [29]. Activation of apoptotic pathways, particularly caspase-dependent pathways, can trigger a cascade of cellular death, resulting in the elimination of cancer cells [27]. Previous studies have shown the induction of G2/M phase cell cycle arrest in various cancer cell lines, attributed to EF24’s interference with cell cycle regulatory systems [30,31]. Furthermore, EF24 has been identified as a potent inhibitor of cancer cell proliferation, selectively targeting essential cellular processes vital for growth and division [30,31,32]. Bertazza et al. revealed EF24’s competence in modulating signaling pathways pivotal in cancer progression, including NF-κB, PI3K/Akt/mTOR, and Wnt/β-catenin pathways, within adrenocortical tumor cell lines SW13 and H295R [33]. Additionally, EF24’s antiangiogenic properties have been demonstrated, effectively hampering neovascularization, a process critical for tumor nutrient supply [34]. By targeting angiogenesis, EF24 can deprive tumors of the necessary blood supply, thereby hindering their growth and metastasis [35]. Lee et al.’s findings indicate that EF24 exerts inhibitory effects on cancer cell migration and invasion in cervical cancer cells, which are considered crucial mechanisms in the process of metastasis [36]. EF24 possesses the capacity to suppress the production of key proteins implicated in the dissemination of cancer cells to other parts of the body, such as matrix metalloproteinases (MMPs) [37]. Moreover, it may disrupt communication pathways that facilitate the motility and invasion of cancer cells [37].

Continued research is necessary to fully understand the mechanisms underlying EF24’s anticancer activity and explore its potential as a therapeutic option in cancer treatment.

### 1.6. Chemosensitization Characteristics

EF24 has demonstrated potent chemosensitization capabilities, indicating its ability to augment the effectiveness of certain chemotherapeutic drugs (Figure 2D). It has been reported that EF24 overcomes drug resistance mechanisms in cancer cells, allowing chemotherapeutic agents to regain their effectiveness [38]. According to the findings of Liang et al., EF24 has the ability to combat multidrug resistance by effectively blocking drug efflux pumps, such as P-glycoprotein (P-gp), which aggressively remove medications from cancer cells [38]. This inhibition translates to increased intracellular accumulation of chemotherapeutic agents, thereby amplifying their lethal effects [38]. The same study also revealed EF24’s synergy with sorafenib, a multikinase inhibitor renowned for its antiangiogenic and antiproliferative effects, setting a new benchmark for advanced hepatocellular carcinoma (HCC) treatment. Actually, by targeting hypoxia-inducible factor 1-alpha (HIF-1α) via cytoplasmic sequestration and subsequent up-regulation of the von Hippel–Lindau tumor suppressor (VHL), EF24 addresses HCC resistance and establishes a breakthrough approach [38]. In addition, Liang et al.’s (2011) findings demonstrated that EF24 possesses the capacity to disrupt DNA repair pathways, increasing the vulnerability of cancer cells to the DNA-damaging impacts of chemotherapeutic drugs. EF24’s effectiveness extends across various HCC subtypes, including PLC/PRF/5, Hep3B, HepG2, SK-HEP-1, and Huh 7 cell lines, as it hampers DNA repair enzymes like poly(ADP-ribose) polymerase (PARP) and the associated damage response pathways [38]. Consequently, EF24 amplifies the cytotoxic influence of DNA-targeting chemotherapeutic drugs [39]. The apoptotic capabilities of EF24, when combined with chemotherapeutic agents, have displayed synergistic effects, bolstering apoptotic signaling cascades and intensifying cancer cell death [38]. By impeding cellular survival pathways, including PI3K/Akt and NF-κB, EF24 contributes to resistance reduction, bolstering susceptibility to chemotherapy’s cytotoxic impacts [38,39].

Chen et al. (2016) found that the modulation of cellular stress responses constitutes an alternative pathway for enhancing chemotherapy via EF24 [23]. EF24 is involved in the enhancement of various stress responses, such as oxidative and endoplasmic reticulum (ER) stress, often provoked by chemotherapeutic treatments. This augmentation extends these stress responses, culminating in increased cytotoxicity and enhanced chemotherapy efficacy [24]. From all the above, the augmentation of cellular apoptosis through EF24 may potentiate chemotherapy when combined.

While further investigation is required to gain a complete understanding of the complex molecular mechanisms and to evaluate the clinical safety and efficacy, it is evident that EF24’s chemosensitization capabilities have been demonstrated and hold significant promise for enhancing cancer therapy.

## 2. Pharmacokinetics of EF24

### 2.1. Absorption of EF24

The pharmacokinetics of EF24 encompass its processes of absorption, distribution, metabolism, and excretion within the human body. Despite the limited availability of comprehensive pharmacokinetic data for EF24, its profile can be influenced by various conditions. The specific absorption properties of EF24 may display variability depending on its method of administration. EF24 is commonly administered orally, utilizing capsules or tablets as the preferred delivery form. Determining EF24’s oral bioavailability relies on multiple factors, including its solubility, stability, and gastrointestinal absorption [1,13].

The lipophilicity of EF24 could facilitate its absorption through the intestinal epithelium due to the ability of lipophilic substances to easily traverse cell membranes [13,40]. However, several factors might limit EF24’s oral bioavailability, such as poor solubility in water, potential metabolism in the gastrointestinal tract or liver, and the presence of efflux transporters that actively expel EF24 from intestinal cells [13,41].

EF24 can also be administered parenterally, including via intravenous (IV) or intraperitoneal (IP) injection [42]. IV injection directly introduces EF24 into the systemic circulation, resulting in rapid and thorough absorption [13,42,43]. The process of IP administration entails the introduction of EF24 into the peritoneal cavity, facilitating its absorption into the circulation through the blood vessels present in the peritoneum [1,42].

A study on the topical administration of EF24, particularly in dermatological applications, remains limited. The transdermal absorption of EF24 may be influenced by factors such as lipophilicity, molecular size, formulation, and skin barrier integrity. Enhancers or specific formulation techniques might augment the transdermal absorption of EF24.

Further research and specific pharmacokinetic studies are necessary to provide detailed information on the absorption of EF24, including its oral bioavailability, absorption rates, and the factors influencing its absorption profile.

### 2.2. Distribution of EF24

Once absorbed into the bloodstream, EF24 is transported to different tissues and organs throughout the body, guided by blood perfusion patterns [1]. Tissues with high blood flow rates, such as the heart, liver, kidneys, and brain, may exhibit comparatively elevated concentrations of EF24 [13]. However, the specific distribution pattern depends on several factors, including tissue perfusion rates, the presence of efflux transporters, and EF24’s affinity for different tissues [13]. EF24 can also bind to plasma proteins, particularly albumin, which affects its distribution [13]. The extent of protein binding influences the availability of EF24 for distribution and the equilibrium between plasma and tissues.

When considering potential therapeutic applications of EF24 for neurological disorders, its lipophilic properties, compact size, and ability to cross the blood–brain barrier (BBB) are significant considerations [44]. These factors play a crucial role in evaluating its suitability for treating neurological conditions.

To provide a more comprehensive understanding of EF24’s distribution patterns, further investigation is essential. Parameters such as tissue-to-plasma concentration ratios, volume of distribution, and extent of absorption in specific tissues should be examined. The combination of pharmacokinetic research and modern imaging techniques can enhance our comprehension of how EF24’s distribution profile varies across different physiological and pathological scenarios.

### 2.3. Metabolism of EF24

EF24 undergoes significant metabolic transformations through enzymatic pathways, primarily in the liver [1,11,40]. However, the specific metabolic pathways and associated enzymes involved in EF24’s metabolism have not been elucidated yet. There is potential for EF24 to undergo a phase I metabolism, followed by subsequent phase II conjugation events [13].

Phase I metabolism encompasses a series of enzymatic processes, mainly involving oxidation, reduction, and hydrolysis, which introduce or unmask functional groups in the molecule [13]. The same study conducted by Reid et al. (2014) revealed that EF24 exhibited a higher metabolic rate in human liver microsomes compared to mice, both tested under similar protein concentrations [13]. This group demonstrated that the similarity in metabolic activity between microsomes from untreated mice and those treated with phenobarbital suggests that the inducible mouse P450s and their human orthologs, specifically CYP3A and CYP2B, do not significantly contribute to EF24 metabolism. However, microsomes obtained from mice treated with 3MC demonstrated a higher metabolic rate, indicating that CYP1A isoforms are responsible for catalyzing EF24 hydroxylation in both mice and humans [13].

Phase II metabolism involves a series of conjugation processes, where the metabolite generated during phase I is combined with endogenous molecules to enhance water solubility and facilitate its elimination from the body. Common conjugation processes in phase II metabolism include glucuronidation, sulfation, and glutathione (GST) conjugation [13]. The exact phase II metabolic pathways involved in EF24’s metabolism remain incompletely defined, necessitating further investigation to identify the specific conjugation processes undergone by EF24.

The clearance of EF24 from the body can affect its overall pharmacokinetics and therapeutic efficacy. Nonetheless, more research is imperative to gain a comprehensive understanding of the precise metabolic pathways, relevant enzymes, and pharmacokinetic implications of EF24 metabolism.

### 2.4. Elimination of EF24

The elimination half-life of EF24 exhibits variability and is influenced by parameters such as dosage and method of administration [13]. Elimination from the human body primarily takes place via renal excretion, predominantly through the urinary system. Metabolites resulting from phase II conjugation processes, such as glucuronides or sulfates, are water soluble and can be excreted as they are or further processed through renal transporters [13]. Reid et al. determined the terminal elimination half-life of EF24 in mice to be 73.6 min. with a plasma clearance value of 0.482 L/min/kg [13]. Biliary excretion is another potential route for EF24 and its metabolites. This process eliminates lipophilic substances like EF24 from the gastrointestinal tract. Subsequently, these substances could be excreted through feces or undergo enterohepatic circulation, wherein they are reabsorbed into the bloodstream, undergo additional metabolism, and are then excreted once again [13]. In liver microsomal preparations, EF24 metabolism generates multiple metabolites aligned with EF24 hydroxylation and reduction processes [13]. EF24 might undergo further metabolic transformations in various tissues before excretion, potentially within organs like the liver or specific target tissues [13].

Further investigation is necessary to gain a comprehensive understanding of EF24’s precise pharmacokinetic properties. Specifically, detailed exploration is required regarding its bioavailability, volume of distribution, elimination pathways, extent of renal and biliary excretion, potential involvement of alternative routes, and the role of specific transporters and enzymes in its elimination process. Additionally, the pharmacokinetics of EF24 could be influenced by various variables, such as drug interactions, individual patient characteristics, and underlying medical conditions. Understanding the pharmacokinetics of EF24 is essential for optimizing dosage regimens, assessing therapeutic efficacy, and evaluating potential drug interactions.

### 2.5. Cytotoxicity of EF24

Understanding the toxicity profile of EF24 is essential to ensure safe and effective cancer treatment [45]. EF24 has shown promise as a cytotoxic agent, with the ability to reduce cancer cell viability, primarily through mechanisms involving caspases and modulation of ROS production [45,46]. However, its effects vary across different cancer cell lines, suggesting a complex interplay of signaling pathways [45]. Notably, the study of Monroe et al. (2011) pointed out that combining EF24 with cisplatin does not mitigate its effects on noncancerous cells but, rather, amplifies them, emphasizing the intricate and sometimes unexpected nature of drug interactions [45]. In addition, recent studies have uncovered diverse ROS responses in different cell lines, adding to the complexity of EF24’s mechanisms [7,45,46]. Furthermore, the ability of EF24 to induce apoptosis in cancer cells represents an important aspect of its toxicity [46]. EF24’s pro-apoptotic effects are thought to be mediated through multiple mechanisms. One key mechanism involves the modulation of various signaling pathways, including those related to cell survival and proliferation [46]. Hence, EF24 holds significant promise as a cytotoxic agent for cancer treatment, but its application is marked by the complex interplay of signaling pathways and diverse effects on various cell lines. A deeper understanding of its impact on both cancer and noncancer cells is essential to ensure its safe and effective use in cancer therapy. When it comes to systemic toxicity, the robust bioavailability of EF24 following intraperitoneal administration facilitates its efficient attainment of the necessary therapeutic concentration, essential for both animal models and clinical applications [47]. Also, the high clearance, surpassing the liver blood flow rate, and the substantial volume of distribution imply rapid metabolism and widespread tissue distribution following intravenous administration of EF24. According to Mosley et al. (2007), EF24 demonstrated no toxicity at doses up to 100 mg/kg, well below the established maximum tolerated dose (MTD) of 400 mg/kg [47]. This finding highlights the superior safety profile of this curcuminoid derivative compared to cisplatin, which has an MTD of 10 mg/kg. Furthermore, post-mortem examinations of EF24-treated animals revealed no signs of damage to the liver, kidney, or spleen in the sacrificed animals [47]. In the same direction, the study of Reid et al. (2014) suggests that EF24 metabolism involves cytochrome P450 enzymes, with species-dependent differences in the predominant metabolic routes, pointing out that reduction is a predominant route of EF24 metabolism [13]. Hence, EF24 emerges as a promising lead compound, demonstrating increased antitumor efficacy in both in vitro and in vivo settings compared to CUR, with preliminary assessments indicating minimal to no observable toxicity [13,47].

Overall, this study seeks to provide a comprehensive overview of EF24, from its structural attributes to its pharmacokinetics and its therapeutic potential in addressing inflammation and cancer. The gathered knowledge can lay down the basis for future research, potential clinical applications, and the development of novel therapies based on EF24.

## 3. Antitumorigenic Effects and Mechanisms

### 3.1. Adrenocortical Carcinoma

EF24 was initially described by Adams et al. (2004) and since then has been investigated in preclinical models [48]. Bertazza et al. explored the use of EF24 in two adrenocortical tumor cell lines, SW13 and H295R [33]. This study revealed a concentration- and time-dependent reduction in cell viability in both cell lines after EF24 treatment. The IC_50_ values for the SW13 and H295R cells were determined to be 6.5 µM and 5 µM, respectively [33]. Additionally, EF24 was found to induce cell death in both cell lines while maintaining a largely unaltered cell cycle. The same study also demonstrated alterations in the PI3k/Akt, MAPK, and Wnt/β-catenin pathways after EF24 treatment [33]. The fact that β-catenin inhibition can influence adrenocortical carcinoma (ACC) cell survival is in agreement with a previous preclinical study reporting nuclear accumulation of β-catenin in over 60% of this type of tumor [49]. In SW13 cells, EF24 was found to increase the phosphorylation of extracellular signal-regulated protein kinases 1 and 2 (Erk1/2), indicating that Erk1/2 may have pro-apoptotic functions under specific circumstances [50,51]. When SW13 adrenal cells were treated with EF24, phospho-β-catenin levels were reduced, while phospho-NF-κB levels increased, suggesting potential crosstalk between the Wnt/β-catenin and inflammation pathways [33].

Bertazza et al. observed an elevation in intracellular ROS levels, a phenomenon previously documented in human breast, prostate, and gastric cancer cells but not in ACC [33]. When exposed to EF24, both SW13 and H295R cells exhibited a significant increase in ROS levels, which could potentially account for the effective antitumor role of EF24 [33]. Actually, EF24 was shown to be capable of generating ROS in MDA-MB-231 (human breast cancer) cells, DU-145 (human prostate cancer) cells, SGC-7901, BGC-823, KATO III (human gastric cancer) cells, and HCT-116, SW-620, and HT-29 (human colon cancer) cells [2,23,26].

Further exploring of the underlying mechanisms by which EF24 affects the PI3k/Akt, MAPK, and Wnt/β-catenin pathways will help with the elucidation of the specific molecular interactions and downstream signaling events that will enhance our understanding of EF24’s mode of action. Understanding the downstream effects of ROS generation and exploring potential strategies to enhance antitumor effects while minimizing oxidative stress-related damage will be crucial in future studies.

### 3.2. Oral Squamous Cell Carcinoma

Oral squamous cell carcinoma (OSCC) represents a prevalent and highly lethal form of cancer [52]. To assess the impact of EF24 on the survival of human oral cancer cells and its influence on the signaling pathways leading to cell death, a study was conducted using CAL-27 cells [52]. The primary objective was to explore the effects of EF24 and cisplatin, a common chemotherapy treatment, across concentrations ranging from 0.1 to 30 μM over a 24 h period [33]. The outcomes of the study revealed that both EF24 and cisplatin led to a significant reduction in CAL-27 cell viability and a decrease in MEK1 and MAPK phosphorylation. Notably, EF24 demonstrated greater efficacy compared to cisplatin [53]. Furthermore, these treatments led to increased levels of phosphorylated caspase-3 and -9, indicative of the initiation of cellular apoptosis. Interestingly, when EF24 was applied simultaneously with phorbol myristate acetate, it potentiated the role of PKC as an indirect regulator of MEK1 and MAPK [53].

While the potential advantages of EF24 in addressing OSCC are promising, the evidence from animal models remains limited. Nonetheless, given the suppression of the crucial MAPK signaling pathway in this specific carcinoma [54], EF24 shows potential as a valuable therapeutic approach for managing OSCC [55].

### 3.3. Nasopharyngeal Carcinoma

The high mortality rates associated with metastatic nasopharyngeal cancer (NPC) underscore the urgent need for effective treatment strategies. EF24, with its enhanced bioavailability and superior anticancer properties compared to CUR, presents potential as a promising therapeutic option [37]. However, there is limited knowledge regarding EF24’s impact on the invasiveness of NPC.

A study conducted by Su et al. has demonstrated EF24’s efficacy in inhibiting the motility and invasiveness of human NPC cells induced by 12-O-Tetradecanoylphorbol-13-acetate (TPA) [37]. The observed inhibitory effects of EF24 were attributed to the reduction of TPA-induced upregulation of matrix metalloproteinase-9 (MMP-9), as a critical mediator of cancer spread [37]. The same authors addressed two distinct mechanisms responsible for downregulating MMP-9 transcription in TPA-activated NPC cells. First, inhibition of nuclear translocation of NF-κB, attributed to suppressed c-Jun N-terminal kinase (JNK) activity, and second, reduction in the interaction between NF-κB and the MMP-9 promoter [37]. Additionally, combining EF24 with a JNK inhibitor (JNK-IN-8) caused an additional decrease in invasiveness and MMP-9 activity induced by TPA in NPC cells, indicating a potential combination treatment strategy [37].

Consequently, EF24 shows promise in inhibiting NPC cell invasiveness by downregulating MMP-9 expression, thus limiting its availability. This positions EF24 as a potential treatment option for NPC, either alone or in combination with other medications.

### 3.4. Breast Cancer

Sun et al. [3] highlighted the cytotoxic potential of EF24 in breast cancer cells. The redox-dependent mechanisms suggest that EF24 induces cytotoxicity by modulating cellular redox balance [3]. Activation of Nrf2 and modulation of redox-sensitive pathways provide insights into the molecular mechanisms underlying EF24’s anticancer effects [3]. Identifying potential resistance mechanisms, such as altered redox signaling pathways or activation of alternative survival pathways, is necessary to develop strategies to overcome resistance. Understanding the specific molecular subtypes or biomarkers that may predict response to EF24 will enable tailored treatment strategies to improve patient outcomes.

Further, EF24 was found to inhibit the proliferation and invasion of tumor necrosis factor (TNF)-bearing cells by reducing the expression of long noncoding RNA (lncRNA) human leukocyte antigen complex group 11 (lnHCG11) and downregulating transcription factor Sp1 expression [56]. EF24 is considered a top CUR analogue candidate, showing a significantly lower IC_50_ value than CUR in solid tumor cells such as cervical and breast cancer cells [56]. Another research study has indicated that EF24 targets noncoding RNAs, such as microRNA-33b (miR-33b) and high mobility group AT-hook 2 (HMGA2), to exert its antitumor effects [57]. However, the involvement of lncRNAs in EF24’s antitumor effects was first demonstrated by Duan et al. (2022), while the association between the N6-methyladenosine (m6A) modification (a common RNA modification in eukaryotic cancer cells) and EF24 was studied by Lan et al. (2021) [56,58]. Actually, Lan et al. (2021) reported that m6A methyltransferase methyltransferase like 3 (METTL3) enzyme is downregulated in triple-negative breast cancer (TNBC) tissue [58]. EF24 may also modulate dysregulated m6A levels by downregulation of HCG11 expression in TNBC cells [56]. Silencing HCG11 was found to cause reduced proliferation and invasiveness in TNBC cell lines, whereas HCG11 expression counteracted the suppressive effect of EF24 on cell proliferation and invasiveness [56]. Further, Sp1 was identified by the same authors as a downstream gene of HCG11 [56]. Sp1 is responsible for the transcription of numerous genes involved in the onset and development of various cancers [59]. It was confirmed that HCG11 directly binds to Sp1 in TNBC cells [56]. In addition, the involvement of mithramycin A (a gene-selective Sp1 inhibitor) slowed the growth of TNBC xenografts in vivo by preventing Sp1 from boosting the transcription of MMP2 [60]. Furthermore, HCG11 reduced Sp1 ubiquitination and was reported to increase Sp1 protein levels in TNBC [56]. The ubiquitin–proteasome system plays a crucial role in protein degradation, and previous studies have demonstrated the ability of lncRNAs to modulate downstream gene ubiquitination at the post-translational level [61,62,63].

The exact mechanism by which EF24 influences Sp1 ubiquitination is not fully understood and requires further investigation. It is possible that EF24 directly or indirectly affects the enzymes involved in the ubiquitination process or modulates the stability of proteins that regulate Sp1 ubiquitination. Understanding the relationship between EF24 and Sp1 ubiquitination is important because Sp1 is implicated in the transcription of genes involved in cancer progression. By modulating Sp1 ubiquitination, EF24 may influence the expression of these genes and potentially affect tumor growth and metastasis.

The research conducted by Davis et al. (2002) showed a strong affinity between estrogen and protein disulfide isomerase (PDI), which is present in several tissues and cancer cell lines [64]. Under conditions of nitrosative stress, PDI modulates the ratio of estrogen receptors (ER) alpha and beta by exerting differential control over the amounts of these two proteins [19,64]. However, the compound EF24 has demonstrated the potential to mitigate this effect by preserving the chemical integrity of PDI under conditions of nitrosative stress. In general, EF24 serves to safeguard PDI from nitrosative stress by eliminating free radicals, which is crucial for maintaining the equilibrium of the ERβ/ERα ratio. This discovery provides insights into regulating estrogenic status and emphasizes the potential of EF24 as a therapeutic agent in conditions associated with PDI dysfunction caused by excessive ROS production.

### 3.5. Lung Cancer

Lung cancer exhibits a diverse array of forms and is among the most prevalent and lethal malignancies [15]. Although chemotherapy demonstrates efficacy in early-stage treatment, its severe adverse effects on normal cells and the need for high doses present significant challenges. Addressing these concerns, EF24 has emerged as a promising candidate for lung cancer treatment. In the same study conducted by Chang et al., the application of EF24 at concentrations of 0.5-4 μM for 24 and 48 h significantly inhibited the proliferation and colony formation of several human non-small cell lung cancer (NSCLC) cell lines [15]. Immunofluorescence analysis of treated cells revealed increased expression of the transmembrane glycoprotein FAS (a cell death receptor). Furthermore, the same authors reported an upregulation of the apoptotic regulatory proteins, including cleaved caspase-3, cytochrome c, and apoptosis regulator BAX. These findings strongly suggest that EF24 induces programmed cell death in cancer cells, as illustrated in Figure 3A. Verification of EF24’s tumor-cytotoxic and antiproliferative properties was achieved through in vivo tests on immunodeficient mice bearing NSCLC xenografts [15]. Treated animals exhibited reduced levels of the proliferative marker Ki-67 protein, smaller tumor xenografts in terms of both size and weight, and no discernible pathological changes in vital organs such as the spleen, liver, kidney, and heart [15]. EF24 administration was found to induce the generation of ROS in a dose-dependent manner. Additionally, EF24 treatment initiated the formation of autophagosomes and led to an increased abundance of shortened mitochondria with abnormally low cristae in NSCLC cells [15]. These results point to mitochondria-mediated programmed cell death as the primary mechanism driving EF24’s selective toxicity towards tumors, facilitated by excessive ROS generation. This concept is supported by evidence indicating that cancer cell survival rates notably increase when they are pre-treated with ROS scavengers like catalase and N-acetyl-L-cysteine (NAC) prior to EF24 administration [15].

The study conducted by Onen et al. (2015) investigated the effects of EF24, both alone and in combination with cisplatin and oxaliplatin, on malignant pleural mesothelioma malignancy (MSTO-211H) and nonmalignant mesothelial (Met-5A) cell lines [65]. The findings from this study revealed that EF24 exhibited cytotoxic effects on cancer cells at doses exceeding 2 μM, surpassing the toxic impacts of the tested platinum-based compounds used individually. Additionally, pretreatment with EF24 enhanced cancer cell susceptibility to chemotherapy. Administering EF24 before oxaliplatin treatment amplified DNA fragmentation, increased cleaved caspase-3 protein levels, and downregulated antiapoptotic genes BCL2L1 and BCL2, ultimately facilitating apoptosis [65]. Conversely, nonmalignant cells pretreated with EF24 displayed increased resistance to the deleterious effects of cisplatin and oxaliplatin when compared to nonpretreated MET-5A cells [65]. In a separate study by Kasinski et al. [66], brief exposure of NSCLC A549 cells to 5 μM EF24 was found to effectively inhibit the phosphorylation and subsequent degradation of the inhibitor of κB kinase (IκB). Consequently, the intact IκB hindered NF-κB translocation to the cell nucleus, impeding the antiapoptotic processes of cancer cells [66]. Moreover, N-substituted modifications to EF24 yielded promising outcomes. Notably, a specific analog, 13d (depicted in Figure 4A), exhibited potent suppression of the NF-κB survival pathway in lung cancer cell lines [67]. This led to pyroptosis, an inflammation-induced type of cell death (as depicted in Figure 3D). Furthermore, compound 13d induced cell cycle arrest in the G2/M phase solely in malignant cells, while normal cell survival remained unaffected [67].

Furthermore, Wu and colleagues (2017) synthesized twenty distinct asymmetric analogs of EF24 and assessed their antitumor efficacy in three lung cancer cell lines: A549, LLC, and H1650 [68]. Most evaluated substances displayed significant efficacy in inhibiting cancer cell viability and colony formation. Among these, compound 81 (illustrated in Figure 4B) exhibited remarkable cytotoxic properties, with IC_50_ values ranging from 6.1 to 6.8 μM across various cell lines. Compound 81 also demonstrated dose-dependent inhibition of cancer cell migration. Subsequent investigation revealed that compound 81 induced apoptosis in A549 cells by hindering IκB degradation, enhancing ROS production, and activating the c-Jun N-terminal kinase (JNK) pathway [68].

EF24’s effectiveness in suppressing cancer cell proliferation, promoting programmed cell death, and initiating mitochondria-mediated cell death processes positions it as an attractive candidate for further exploration and advancement in the treatment of lung cancer therapy.

### 3.6. Hepatocellular Carcinoma

Hepatocellular carcinoma (HCC), a significant contributor to global cancer-related mortality, is particularly prevalent in Asian regions [69]. Current systemic treatments for HCC exhibit limited effectiveness due to drug resistance development and adverse impacts on healthy tissues [69]. However, recent research focused on EF24 demonstrates its potential in suppressing hepatic carcinoma cell proliferation, both in controlled laboratory settings (in vitro) and living organisms (in vivo), without affecting noncancerous liver cells [26]. The same study emphasized EF24’s ability to inhibit liver cancer cell proliferation by downregulating the NF-κB pathway [26]. EF24 induces cell cycle arrest in the G2/M phase in various cancer cells, including liver cancer cells. This effect involves decreased expression of cyclin B1 and Cdc2, along with elevated levels of tumor protein (p53) and cyclin-dependent kinase inhibitor (p21), as depicted in Figure 5 [70,71]. Furthermore, EF24 reduces ERK phosphorylation and downregulates phospho-Akt, suggesting its potential role in suppressing cellular survival pathways [72]. Beyond these attributes, EF24 exhibits antiangiogenic effects by inhibiting vascular endothelial growth factor (VEGF) production and vascular endothelial cell proliferation [20].

The study by Liang et al. (2013) investigates the effects of continuous sorafenib treatment on intratumoral hypoxia in HCC patients and animal models [38]. Sorafenib administration initially reduces microvessel density, increases HIF-1α protein levels, and enhances its activity [38]. In addition, the same group has found that moderate sorafenib doses effectively suppress tumor growth, while hypoxia leads to resistance by hindering growth inhibition and apoptosis [38]. Hypoxia triggers the stabilization of HIF-1α, resulting in the upregulation of target genes like multidrug resistance-1 (MDR1), glucose transporter-1 (GLUT-1), and VEGF, associated with NF-kB activation and antiapoptotic gene upregulation [38]. The same authors have found that silencing HIF-1α sensitizes HCC cells to sorafenib, suggesting that hypoxia-induced HIF-1α correlates with sorafenib resistance (Figure 6A). Based on this, the study proposes combining sorafenib with an HIF-1α inhibitor as a therapeutic approach [38]. Additionally, they proposed that EF24 synergistically reduces cell viability and promotes apoptosis in concern with sorafenib, especially in conditions of hypoxia [38]. In murine models, the EF24–sorafenib combination suppresses tumor growth and metastasis, countering hypoxia-induced resistance and angiogenesis.

Numerous studies have highlighted EF24’s antiproliferative properties; however, its impact on cancer metastasis has not been well-studied. The study of Zhao et al. (2016) presented the initial evidence of EF24’s inhibition of migration and invasion in HCC cells [73]. Cellular migration and invasion are pivotal in metastasis development. EF24 has been shown to curb migratory capacity and diminish filopodia formation in HCC cells, consistent with CUR’s reported effects on cancer stem-like cells and EF24′s effects on the microtubule cytoskeleton [74,75].

Src, a nonreceptor tyrosine kinase, plays a key role in HCC metastasis, mediating processes like proliferation, migration, and invasion [73]. Elevated phosphorylated Src (p-Y416Src) levels correlate with lymph node metastasis in HCC [75,76]. Zhao et al.’s study explored Src expression and activation in primary HCC cells and the corresponding metastatic lymph nodes, unveiling increased p-Y416Src in the nodes compared to original HCC tissues [73]. In other contexts, CUR modulates the Src–Akt axis in bladder cancer, and EF24 inhibits migration and the epithelial–mesenchymal transition (EMT) in melanoma cells by suppressing Src, as established by Charpentier et al. (2014) [77]. Zhao et al. [73] delved into EF24’s impact on Src expression and phosphorylation in HCC cells, finding that EF24 effectively reduces Src phosphorylation without altering its overall level (Figure 6A). This suggests that targeting Src could mitigate HCC metastasis, wherein EF24’s suppression of Src phosphorylation appears to be a promising approach.

Considering that HCC is a heterogeneous disease with diverse underlying causes and genetic alterations, stratification based on molecular profiles and biomarkers could help in the identification of subsets of potential patients as beneficiaries of EF24 treatment. Personalized medical approaches may be necessary to optimize treatment outcomes for individual patients. Resistance to therapeutic agents is a significant challenge in cancer treatment. Elucidating the mechanisms of resistance to EF24 and developing strategies to overcome or prevent resistance is crucial for long-term treatment success.

### 3.7. Gastric Cancer

The research conducted by Zou et al. (2016) has provided significant insights into the impact of EF24 treatment on various aspects of human gastric cancer cells [22]. Their study unveiled that EF24 treatment effectively deactivates thioredoxin receptor 1 (TrxR1), triggers endoplasmic reticulum (ER) stress, and selectively induces apoptosis in these cancer cells [22]. The relevance of the thioredoxin system, comprising Trx1 and TrxR1, has been underscored in cancer chemotherapy due to their increased expression in different human cancer types, associated with augmented tumor growth, drug resistance, and unfavorable prognoses (Figure 6B) [22]. On the other hand, the investigation also proposed that the glutathione (GSH) system could act as a reservoir for the thioredoxin system [22]. Hence, combination therapies that inhibit both GSH and TrxR antioxidant pathways may yield successful outcomes [22]. A more comprehensive exploration of additional antioxidants, their individual impacts, and their combined effects could provide crucial insights for enhancing cancer treatment strategies.

The ER assumes a pivotal role in modulating cellular responses to stress, and disturbances in ER equilibrium can trigger the unfolded protein response [78]. The accumulation of misfolded proteins within the ER leads to ER stress, ultimately culminating in apoptosis [78]. In line with these findings, EF24 treatment concomitantly elicits an ER stress response, evident through increased levels of phosphorylated α-subunit of eukaryotic initiation factor 2 (p-eIF2α), activated transcription factor 4 (ATF4), and increased quantities of C/EBP-homologous protein (CHOP) [79]. Elevated CHOP expression can result in cell cycle arrest, prompting cell apoptosis [79]. Furthermore, CHOP-mediated apoptosis can induce cell death by suppressing the expression of the cell cycle regulatory protein p21 [79]. Zou et al. (2016) demonstrated that siRNA-mediated knockdown of CHOP moderately inhibits EF24-induced apoptosis, implying that EF24-triggered ER stress is a secondary response to ROS generation caused by EF24 in gastric cancer cells [22]. The same study emphasizes the connection between EF24-induced oxidative stress and ER stress-based “Unfolded Protein Response” (UPR), which can amplify the lethal repercussions of EF24 in gastric cancer cells [22]. Exploiting ER stress-induced apoptosis in cancer cells stands as a significant target for the development of antineoplastic drugs.

The fact that EF24 targets TrxR1 both in vitro and in vivo and that its cytotoxic effects synergize with 5-fluorouracil (5-FU) opens innovative possibilities for tumor treatment using EF24 in combination with established anticancer drugs or oxidative stress-inducing treatments. This innovative targeting mechanism may lead to the development of potent small molecules (TrxR1 inhibitors) as promising chemotherapeutic agents.

An alternative therapeutic approach related to gastric cancer involves targeting dysregulated proteins within the mTOR pathway [79]. The strategy of targeted therapy has gained attention in gastric cancer treatment, wherein MK-2206 emerges as a potent and selective allosteric inhibitor of the PI3K–Akt–mTOR signaling cascade [79]. The study of Chen et al. (2017) found that MK-2206 effectively inhibits Akt phosphorylation in gastric cancer cells, even at concentrations as low as 0.1 μM [24]. However, a noticeable increase in cell death was only observed at higher concentrations, encompassing both gastric cancer and normal gastric cells. This outcome implies that inhibiting Akt phosphorylation and its activity might not be solely sufficient to provoke cellular death [24]. Addressing this challenge, a potential solution emerges in the form of combining MK-2206 with other cancer therapeutics. Notably, in breast and lung cancer cell lines, the addition of MK-2206 potentiated the impacts of diverse chemotherapeutic and targeted agents, demonstrating the viability of this combinatory approach for gastric cancer treatment [24]. The same study has shown that MK-2206 induces escalating ROS levels in a concentration-dependent manner, spanning from 10 to 40 μM [24,80]. While the mechanism underlying this increased ROS production remains obscure, it was deduced that MK-2206’s ROS-inducing capacity is not dependent on Akt. Genetic silencing-induced Akt knockdown failed to prevent ROS generation within cancer cells, underscoring that MK-2206 triggers ROS via alternative targets and signaling pathways [24]. However, combining therapy has exhibited superior efficacy, particularly when coupled with low-dose ROS inducers, fostering substantial ROS production and instigating cellular apoptosis [81,82]. Both MK-2206 and EF24 contribute to ROS elevation within gastric cancer cells, subsequently triggering ER stress, mitochondrial dysfunction, and apoptosis [81,82].

Chen et al. (2016) also investigated the combinatorial effect of EF24 and rapamycin on human gastric cancer [23]. Their results confirm that EF24 increased sensitivity to rapamycin, inducing apoptosis and selective growth inhibition [23]. Actually, rapamycin led to dose- and time-dependent ROS accumulation, intensified by EF24’s ROS-inducing action [23]. Their combined regimen triggered ER stress pathways, increasing phosphorylated ER R-like protein kinase (PERK), eIF2α, ATF4, and ER stress-related CHOP expression [23,79]. The reduction of ROS via N-acetylcysteine (NAC) and catalase intervention decreases ER stress activation, indicating that rapamycin and EF24’s cooperative anticancer effect involves an ROS-dependent ER apoptotic stress pathway.

The interplay between oxidative stress, ER stress, and mitochondrial dysfunction lies in the synergistic anticancer effects of rapamycin and EF24 in gastric cancer cells. Combining these two agents emerges as a potential therapeutic strategy for improving treatment outcomes in gastric cancer.

### 3.8. Colon Cancer

EF24 exhibited potent anticancer activity by inducing apoptosis and G2/M cell cycle arrest in human colorectal cancer cells (HCCC) [25]. The mechanism of action involved intracellular ROS elevation, particularly H_2_O_2_. Increased ROS levels led to oxidative stress, causing mitochondrial dysfunction, loss of mitochondrial membrane potential (ΔΨ_m_), cytochrome c leakage, and disrupted mitochondrial structure. This mitochondrial injury was accompanied by a decrease in the antiapoptotic protein Bcl-2 and activation of the caspase cascade, ultimately leading to apoptosis (Figure 3B) [25].

EF24 also suppresses cell cycle progression at the G2/M phase and suppresses colorectal cancer cell proliferation. This effect was correlated with diminished cell cycle-related proteins: Cdc2, murine double minute 2 (MDM2), and cyclin B1 (Figure 5) [25]. The ROS-dependent G2/M arrest was evidenced in colorectal cancer cells (HCT-116, SW-620, and HT-29) through reduced Sp1 expression due to EF24 treatment [25].

The findings underscored the potential of EF24 as an effective and low-toxicity anticancer agent for colorectal cancer. Future studies should focus on elucidating the exact molecular and cellular mechanisms underlying EF24’s anticancer effects.

In vivo, EF24 reduces tumor growth, volume, and microvessel density, as well as the expression of colon cancer-promoting genes [2]. EF24 exhibits minimal toxicity and targets various molecular pathways involved in colon cancer, such as COX-2, VEGF, and IL-8 [1,10].

EF24 inhibits the proliferation of intestinal cancer cells both in vitro and in vivo without affecting normal fibroblasts, suggesting that it has potential as a therapeutic or chemo-preventive drug for intestinal cancer and other malignancies and inflammatory settings.

### 3.9. Renal Cell Carcinoma

Kidney cancer often presents as asymptomatic [83] with complex clinical manifestations resulting in high metastasis rates and a poor prognosis [84]. Nephrectomy is the main treatment, but metastatic recurrence affects nearly half of patients due to limited post-operative success in reducing metastases [85]. TRAIL (tumor necrosis factor-related apoptosis-inducing ligand) selectively induces cancer cell apoptosis via death receptors 4 and 5 (DR4/DR5) and caspase-8 (Figure 3C) [86,87]. However, primary tumors frequently resist TRAIL-induced apoptosis [88]. Researchers tackled this by combining TRAIL with natural dimethoxycurcumin (DMC) or synthetic (EF24) analogues of CUR [89]. In renal adrenocarcinoma cells, TRAIL combined with DMC or EF24 significantly reduced cell viability (32.73% and 11.39%, respectively), and both analogues hindered cell migration [90]. EF24 and TRAIL combination enhanced DR4 expression, overcoming resistance and decreasing viability, whereas DMC has distinct cytotoxic mechanisms (Figure 3C) [88].

Elevated ROS production correlates with increased matrix metalloproteinase (MMP)-2 and -9 expression, promoting cell invasion and metastasis in colorectal cancer [89]. H_2_O_2_/TRAIL-induced death has been observed in TRAIL-resistant cells [91]. Combined EF24 and TRAIL treatment causes increased peroxidase activity by 555% after 24 h and 302% after 72 h, suggesting reduced metastatic potential [16]. While increased H_2_O_2_ could increase metastatic risk, EF24’s peroxidase activation lowered H_2_O_2_ levels, reducing MMP-2 and MMP-9 expression and hindering ACHN cell migration (Figure 3C) [16].

EF24 shows promise as a therapeutic agent against highly resistant and frequently metastasizing renal adenocarcinoma cells. The exploration of EF24’s potential and its synergistic effects with other treatments may offer new avenues for the management of kidney cancer and improved patient outcomes.

### 3.10. Prostate Cancer

Yang et al. (2013) conducted a study demonstrating EF24’s robust anticancer potential in human prostate carcinoma DU145 cells and immunodeficient mouse tumor xenografts [27]. Their findings revealed that a 5 μM EF24 treatment for 24 h induced apoptosis by inhibiting the NF-κB pathway, irrespective of interferon-alpha (IF-α)-triggered NF-κB activation [27]. Notably, EF24 spared the signal transducer and activator of the transcription (STAT3) signaling pathway [27]. Basically, the study was focused on miRNA modulation linked to cancer cell survival. EF24 led to a significant reduction in pro-oncogenic miRNA-21 in DU145 cells as well as miRNA-23 downregulation in both cells and xenografts. Additionally, EF24 elevated tumor-suppressive miRNAs (miR-10a, miR-206, miR-345, and miR-409) and the genes Programmed Cell Death 4 (PCD4) and tumor suppressor PTEN, contributing to its anticancer effects (Figure 7) [27]. A 4-week EF24 treatment significantly reduced the size of xenografted cancer formations in nude mice, reinforcing its potential as a prostate carcinoma therapeutic [27].

The findings on EF24’s anticancer potential in prostate carcinoma cells and mouse tumor xenografts provide a strong foundation for future research and clinical development. Based on the achieved results, researchers can work towards the development of EF24-based therapies that offer improved outcomes for prostate carcinoma patients.

### 3.11. Thyroid Carcinoma

Parafollicular C cells in the thyroid gland have a crucial role in calcitonin secretion [92]. However, in rare cases, they can lead to a neuroendocrine tumor known as medullary thyroid cancer (MTC). A study conducted by Bertazza et al. (2018) explored EF24’s effects on human MTC cell cultures [92]. EF24 concentrations from 0.5 to 100 μM were tested on TT and MZ-CRC-1 cell lines using sulforhodamine B (SRB) and 3-(4,5-dimethylthiazol-2-yl)-2,5-diphenyl tetrazolium bromide (MTT) assays [92]. TT cells had a mean IC_50_ of 4 μM for EF24, while the IC_50_ for EF24 in MZ-CRC-1 cells was 6.55 μM. The effect of EF24 was also tested with the PI3K inhibitor ZSTK474 and the kinase inhibitor XL184. EF24+ZSTK474 weakly synergized, while EF24+XL184 showed a strong synergy. Moreover, EF24 raised ROS by 65% in TT and 29% in MZ-CRC-1 cells [92]. EF24 reduced calcitonin secretion in both cell types, with a slight impact on migration as a crucial step for tumor growth, hinting that EF24’s effect on MTC viability might relate to PI3K inhibition and enhanced ROS generation [92].

EF24 shows potential as a second-line therapy for MTC. Combined with other agents, it exhibits synergistic effects, offering promise for improved patient outcomes. Further research and clinical studies are needed to fully validate EF24’s therapeutic value in managing MTC.

### 3.12. Ovarian Cancer

Ovarian cancer is one of the most frequent gynecological malignant causes of death, affecting thousands of women worldwide every year [93]. To address this, Tan et al. (2010) explored EF24’s impact on ovarian cancer cells, investigating platinum-sensitive IGROV1 and platinum-resistant SK-OV-3 cell lines [20]. The results obtained by the same study showed that EF24 inhibited cell proliferation, with IC_50_ values of 1.6 μM for IGROV1 and 2.4 μM for the SK-OV-3 cell line after 24 h [20]. Elevated cleaved caspase-3 and reduced BCL-2 levels confirmed pro-apoptotic mechanisms triggered by EF24 (Figure 3A). Intriguingly, pretreatment with EF24 enhanced platinum-induced apoptosis in cisplatin-resistant cells [20]. In addition, VEGF, as one of the pivotal molecules in tumor angiogenesis and ovarian cancer prognosis, showed reduced secretion in IGROV1 cells treated with EF24, suggesting possible posttranscriptional regulation [94].

The study conducted by Tan et al. (2010) reported that EF24 exhibited antioxidant properties, lowering ROS generation even after H_2_O_2_ addition [20]. This is in contrast with other studies reporting increased ROS after EF24 application, which might be due to the specific experimental conditions [20]. Actually, reductive stress induced by low ROS concentrations might contribute to EF24’s cytotoxicity in IGROV1 cells. EF24’s dose-dependent cytotoxic effect was reported in cisplatin-resistant A2780 ovarian carcinoma cells [30]. EF24 caused G2/M phase arrest in A2780 cells and reduced the levels of key cell cycle regulators, including CycA, CycB1, CycD1, Cdc2, and Cdc25 (Figure 5). Additionally, EF24 activated the tumor suppressor protein p53 in a time-dependent manner, leading to apoptosis through caspases, the Fas receptor, and PTEN pathways [30]. EF24 also prevented PTEN degradation, thereby enhancing pro-apoptotic mechanisms and preserving the integrity of p53 [30].

Addressing ovarian cancer’s glycolytic energy production, EF24 was found to reduce glucose uptake and glycolysis rates in A2780, SK-OV-3, and OVCAR-3 cell lines [20,30]. Additionally, EF24 hindered cancer cell attachment, migration, and metastasis, along with reducing the size and number of metastatic formations in tumor xenografts implanted in nude mice [13].

EF24 appears to be a promising antiovarian tumor agent, potentially revolutionizing ovarian cancer treatment strategies.

### 3.13. Osteosarcoma

Osteosarcoma (OS) is a malignant bone tumor originating from mesenchymal cells and represents the most prevalent primary bone cancer, posing fatal risks for both children and adults [95]. Its diverse genotypes and frequently changing genetic profile represent challenges for genetic-based treatment approaches [96]. Basically, chemotherapy remains the foremost anticancer therapy for OS, directly targeting tumor masses and mitigating their harmful effects [96,97]. An emerging candidate in OS research is EF24, with its capacity to regulate pathways such as ROS or NF-κB and to induce apoptosis across different cancer types [98]. Within the context of OS, a study utilizing an OS-derived cell line had dual objectives [99]. EF24 exhibited greater potency than CUR, leading to nuclear condensation and fragmentation (hallmarks of apoptosis) in Saos2 cells. Both EF24 and CUR significantly increased the apoptotic rate of Saos2 cells, with EF24 displaying higher efficacy [99]. Activation of caspase-3/-7 indicated involvement of the apoptotic pathway alongside increased cleaved poly(ADP-ribose) polymerase (PARP) levels. Changes in the expression of regulatory proteins Bcl-2, BAX, and p53 further suggested their engagement in the apoptotic process induced by EF24 and CUR [99]. Both CUR and EF24 induce cell death in Saos2 human OS cells through two distinct pathways: the mitochondria-mediated intrinsic pathway and the death receptor-mediated extrinsic pathway (Figure 3B) [99]. In the mitochondria-mediated pathway, both compounds influence the expression levels of Bcl-2 family proteins, leading to a shift in the balance between pro-survival and pro-apoptotic factors [99]. This imbalance triggers mitochondrial outer membrane permeabilization (MOMP), resulting in the release of cytochrome c and the subsequent activation of caspase-9, ultimately leading to apoptotic cell death [99]. In the death receptor-mediated pathway, both compounds induce the upregulation of FasL, an apoptotic ligand, which leads to the activation of caspase-8, another initiator caspase, further contributing to apoptotic cell death [99].

In a recent study, researchers investigated how EF24 affects ferroptosis in osteosarcoma (OS) cells [29]. EF24 was found to induce ferroptosis by increasing the expression of HMOX1 (heme oxygenase-1) and raising intracellular iron levels (Figure 6C). This modulation of HMOX1 expression and intracellular iron accumulation, as suggested by previous research (Dixon et al., 2012), led to lipid peroxidation and ultimately resulted in ferroptotic cell death (Figure 6C) [100]. Interestingly, OS cells with higher HMOX1 expression were more sensitive to EF24 treatment, suggesting that EF24-mediated ferroptosis could be a promising therapeutic approach for treating OS [29].

EF24’s multifaceted action, encompassing apoptosis and ferroptosis, offers promise for enhancing OS treatment. Collaborative efforts between researchers and medical experts remain essential for advancing understanding and developing effective OS treatments.

### 3.14. Neuroblastoma

Neuroblastoma (NB) is a tumor that arises from precursor cells derived from the neural crest during embryonic development [101]. This condition is thought to result from disturbances in the signaling pathways that control the development of the sympathetic nervous system, including neurotrophin receptors [101]. Additionally, disruptions in specific transcription factors, which play a crucial role in determining the developmental lineage and differentiation, contribute to the complexity of this disorder [101]. Unfortunately, there are currently no well-established curative treatment options available for a significant number of patients with NB.

Only one study addressing the impact of EF24 on NB malignancies has been identified in the existing literature [102]. This extensive study employed various systematic approaches to delve deeper into the molecular signaling pathways associated with EF24’s interaction with NB cells. Another research study underscored the significance of radiotherapy (RT) as a pivotal component of clinical protocols for NB treatment [103]. Nonetheless, RT is not without limitations, including the potential for disease relapse and the emergence of drug/radiation resistance, possibly via alternative molecular signaling routes [103]. A variety of studies probed into these mechanisms, revealing that therapeutic doses of radiation triggered NF-κB signaling in human NB cells [104,105,106]. This activation led to an upsurge in TERT (telomerase reverse transcriptase) transcription, increased telomerase activity, and promoted cancer survival and clonal expansion [107]. To address these challenges, the researchers explored EF24, a pharmacologically safe NF-κB antagonist, as a potential countermeasure against the cellular mechanisms influenced by RT in NB cells [102]. In their study, Madhusoodhanan et al. (2009) harnessed in vitro and in vivo NB models, employing diverse laboratory techniques to dissect the intricate network of cell cycle contributors within NB cells and animal models subjected to RT, both with and without EF24 [106]. They subjected genetically distinct human NB cell lines (SH-SY5Y, IMR-32, SK-PN-DW, MC-IXC, and SK-N-MC) to RT doses (2 Gy at a dose rate of 0.81 Gy/min) and incubated them at 37 °C for varying time intervals [107]. In an in vivo NB model employing athymic NCr-nu/nu nude mice, the researchers validated EF24’s potential as an antitumor agent, both independently and in combination with RT irradiation (2 Gy/D for 5 days/week, totaling 20 Gy). Intratumoral administration of EF24 (200 µg/kg) effectively hindered NB growth in the xenograft model, as evidenced by standard tumor volume measurements and corroborated by [18F]-2-fluoro-2-deoxy-D-glucose (FDG)-positron emission tomography/computed tomography (FDG-PET-CT) imaging [106]. The results obtained from the study of Aravindan et al. (2011) indicated that CUR caused inhibition of RT-induced NF-κB-DNA binding activity and promoter activation [107]. Furthermore, it sustained the suppression of NF-κB by obstructing NF-κB-dependent TNF-α transactivation and intercellular secretion across various genetically diverse human NB cell types, encompassing SH-SY5Y, IMR-32, SK-PNDW, MC-IXC, and SK-N-MC [105]. Moreover, EF24 effectively quelled RT-induced NF-κB-TNFα cross-signaling, which holds a pivotal role in the transactivation and translation of pro-survival proteins like Inhibitors of Apoptotic Proteins 1 and 2 (IAP1, IAP2) and Survivin, ultimately resulting in elevated cell survival [105,107]. Additionally, CUR treatment adeptly curbed RT-induced NF-κB activation, impeding hTERT transactivation, promoter activation, telomerase activation, and subsequent clonal expansion [107,108].

EF24 could be found as a potential therapeutic agent for enhancing the treatment of NB and warrants further exploration in clinical settings. Continued research into EF24’s effects on IR-induced molecular pathways in NB can provide valuable insights and potentially lead to more effective treatments for this challenging cancer.

### 3.15. Leukemia

Acute myeloid leukemia (AML) is the most common leukemia in adults, and current therapy primarily involves chemotherapeutic agents. However, drug resistance and severe side effects necessitate the search for new therapeutic drugs, particularly natural products with lower toxicity [46].

EF24 was found to moderately enhance ROS production and decrease glutathione (GSH) levels, indicating oxidative stress in leukemia cells [46]. Interestingly, there was no increase in oxidized GSH (GSSG) levels, and the ratio of GSH/GSSG remained unchanged [46]. The same authors found that the addition of the ROS scavenger and GSH precursor, NAC, at high concentrations induces prevention of ROS production, GSH depletion, and cell death [46]. Furthermore, catalase prevented ROS production but did not impact GSH depletion or cell death, suggesting a nonoxidative mechanism of EF24-induced apoptosis in leukemia cells [46]. EF24 did not activate the Nrf2 signaling pathway or induce changes in stress-response protein expression [46], which is a major cellular defense mechanism against oxidative stress [109]. The GSH depletion may also be attributed to EF24 forming adducts with GSH and NAC, converting cytotoxic EF24 into noncytotoxic complexes (NAC-EF24 and GSH-EF24) through these interactions [46]. Understanding these interactions is essential, as they may affect EF24’s effective dose and treatment success against AML.

Comparing EF24 to dimethoxycurcumin (DMC), another analogue of CUR, EF24 demonstrated a 10-fold greater effectiveness in HL-60, U937, and MV4-11 (AML cell lines) [110]. Treatment with EF24 increased the apoptotic fraction of sub-G1 in HL-60 cells. Actually, in the concentration range of 0.25–2 µM, EF24 activated caspase-8 and -3 (extrinsic pathway) but did not affect caspase-9 activity (intrinsic pathway) [110]. Thus, EF24’s antileukemic effect was achieved through the extrinsic apoptotic pathway and/or by arresting the cell cycle in the S-phase [110]. The same authors reported that lower EF24 concentrations increased ERK activity, while JNK1/2 or p38 MAPK activity remained unaffected. However, higher EF24 concentrations (>2 µM) inhibited ERK and increased p38 and JNK1/2 activity, leading to caspase-mediated apoptosis (Figure 3D) in HL-60 and MV4-11 cell lines [110]. Additionally, activation of the ERK pathway resulted in drug resistance and reduced apoptosis in leukemia cells [111,112]. At high doses, EF24 decreased the phosphorylation of protein phosphatase 2A (PP2A), a key negative regulator of numerous tumorigenic pathways, thereby upregulating its activity [110]. Consequently, EF24-induced PP2A activity negatively regulated ERK and caused apoptosis in AML cells [110].

EF24 appears to be a novel PP2A-activating drug that, in combination with traditional chemotherapeutics or tyrosine kinase inhibitors, may hold promising potential for drug-resistant AML treatment [110].

### 3.16. Melanoma

Malignant melanoma is the most common cutaneous cancer, known for its high metastatic potential and rapid progression [113]. Unfortunately, once melanoma metastasizes, it becomes resistant to most chemotherapeutics, resulting in a poor prognosis [113]. For effective therapy and a better prognosis in complex diseases like cancer, drugs that target multiple signaling mechanisms are needed. Curcumin (CUR) and its analogues have emerged as promising agents that modulate various cellular pathways involved in melanoma pathogenesis, including mammalian sterile 20-like kinase 1 (MST1), JNK, forkhead box O3 (Foxo3), Bcl-2 interacting mediator of cell death (Bim-1), myeloid leukemia cell differentiation protein (Mcl-1), Bcl-2, Bcl-2-like protein 4 (BAX), and JAK-2/STAT3 [114,115,116,117].

EF24 has shown potential for reducing the motility and the epithelial–mesenchymal transition (EMT) of Lu1205 and A375 melanoma cell lines [118]. It achieves this by upregulating miR-33b expression, which, in turn, leads to decreased expression of high mobility group A2 (HMGA2) at both the protein and mRNA levels (Figure 7) [118]. Additionally, EF24 upregulated the expression of the epithelial cell marker E-cadherin and reduced the expression of the mesenchymal markers vimentin and N-cadherin, promoting epithelial differentiation in melanoma cells [118]. By inducing the overexpression of miR-33b, EF24 modulated HMGA2-dependent cytoskeletal organization, focal adhesion assembly, and activation of focal adhesion kinase (FAK), Src, and small GTPase (RhoA), thereby restricting metastasis [57]. Interestingly, EF24 achieved this without influencing the total expression of FAK, Src, and RhoA but rather attenuated their phosphorylation and GTP coupling [57]. Moreover, EF24 inhibited the phosphorylation of STAT3 [57], a key transcription factor promoting cancer EMT [118]. Hence, one can conclude that EF24 effectively targeted the HMGA2/STAT3/EMT signaling pathway [57].

The recent research on EF24 in melanoma treatment shows promise in targeting multiple signaling pathways involved in melanoma progression and metastasis. Future research efforts should focus on clinical translation, combination therapies, and a deeper understanding of the underlying molecular mechanisms to advance treatment options for melanoma patients.

### 3.17. Antiangiogenic Effects of EF24

The tumor vasculature is crucial for supplying nutrients and supporting the growth and metastasis of tumors [119]. Targeting tumor angiogenesis, the development of blood vessels in solid tumors, is a promising strategy to block solid tumor malignancies [119]. Various approaches, including antiangiogenic agents that interfere with the formation of new blood vessels and vascular targeting agents that disrupt existing tumor vasculature, are under investigation [120]. However, these therapies face challenges due to the distinct properties of tumor blood vessels in solid tumors, which can lead to insufficient blood flow within the tumor [121].

The potent anticancer agent EF24 has been extensively studied for its vasculature-dependent activities. Initial research by Adams et al. (2004) demonstrated promising antiangiogenic properties of EF24 comparable to those of the antiangiogenic drug TNP-470 [48]. In vivo studies showed that EF24 effectively suppressed breast tumor growth with minimal toxicity in a mouse xenograft model [48], emphasizing its potential as a valuable chemotherapeutic agent. While the precise mechanisms of EF24’s antiangiogenic effects are still in the early stages of research, the study conducted by Shoji et al. (2008) revealed an intriguing aspect [35]. It was found that EF24 could be targeted specifically to tissue factor (TF)-expressing tumor-associated vascular endothelial cells (VECs) and tumors by using an active site-inactivating factor fVIIa (FFRck-fVIIa) as a carrier [35]. This targeted delivery of EF24 significantly enhanced its effectiveness as an anticancer and antiangiogenic compound, leading to the inhibition of tumor growth in human breast cancer xenografts in mice [35]. The EF24–FFRck-fVIIa conjugate selectively destroyed TF-expressing VECs and cancer cells, demonstrating a promising strategy for cancer therapy [35]. Another study investigating the effect of EF24 on tumor angiogenesis revealed its remarkable ability to significantly reduce the expression of VEGF and IL-8, which are potent inducers of capillary growth in tumors [34]. Furthermore, EF24 treatment led to a substantial decrease in platelet/endothelial cell adhesion molecule-1 (CD31) staining, indicating the inhibition of tumor angiogenesis [34]. In a valuable zebrafish in vivo model study, EF24 demonstrated antiangiogenic activities on normal blood vessels, resulting in a significant reduction in angiogenic development [122]. Surprisingly, EF24 exerted its antiangiogenic effects by inhibiting NF-κB translocation, even though an NF-κB blockade typically triggers angiogenic signaling. This suggests that EF24’s multitargeting character may contribute to its antiangiogenic activity [122].

The obtained results collectively support the usefulness of EF24 in blood vessel-targeted strategies for cancer treatment. However, addressing the low bioavailability and rapid metabolism of EF24 remains critical for further development.

## 4. EF24 Derivatives and Drug Delivery Systems

### 4.1. EF24 Derivatives

Numerous research groups have been working on developing novel analogues of EF24 to improve anticancer treatments. Lagisetty et al. (2012) created a hydrazinonicotinic acid conjugate with enhanced water solubility and significant antitumor effects [40]. Wu et al. (2017) identified one of the 20 purified EF24 analogues with superior activity against A549 cells, displaying strong antimigration and antiproliferative effects [68]. Xie et al. (2017) designed a series of EF24 analogues, finding one that showed excellent inhibition of both inhibitor of nuclear factor kappa-β kinase subunit beta (IKKβ) activity and pancreatic cancer development [123]. Schmitt et al. (2017) synthesized 14 EF24 analogues, which demonstrated promising antiproliferative activity against eight cancer cell lines and superior antiangiogenic and vascular-disruptive effects [34]. Chen et al. (2018) discovered an EF24 analogue with significantly higher inhibitory activity against IKKβ and the ability to induce apoptosis and cell cycle arrest in multiple cancer cell lines [124].

### 4.2. EF24 Drug Delivery Systems

Improving the effectiveness of chemotherapy drugs and reducing their side effects is a crucial goal in oncology. Specific drug delivery systems have emerged as a promising tool for achieving this objective. One innovative approach involves leveraging tissue factor expression in cancer cells and associating drug delivery systems with tissue factor on the surface of these cells [125]. This strategy has shown enhanced delivery efficacy in human breast and melanoma cell lines [125]. Additionally, the conjugation of EF24 with coagulation factor VIIa (EF24–FFRmk-fVIIa) led to apoptosis in tumor cells and a reduction in tumor size in breast cancer xenografts in athymic nude mice [35]. These targeted drug delivery systems hold tremendous potential for improving therapeutic outcomes while minimizing toxicity.

Another study [126] successfully developed a liposomal formulation of EF24 using a “drug-in-CD-in liposome” approach, as described by Agashe et al. (2011). This innovative method involves combining EF24 with hydroxypropyl-β-cyclodextrin (HPβCD) to enhance its aqueous solubility [126]. The result was the preparation of EF24 liposomes with improved characteristics. Furthermore, in vitro experiments revealed that EF24 liposomes exhibited superior antiproliferative activity compared to plain EF24 when tested on lung adenocarcinoma H441 and prostate cancer PC-3 cells [126]. In vivo studies in rats indicated efficient clearance of Tc-99m-labeled EF24 liposomes from the bloodstream, primarily through uptake in the liver and spleen. This research suggests that the “drug-in-CD-in liposome” approach is a viable strategy for formulating an effective parenteral preparation of EF24 that remains antiproliferative and offers opportunities for biodistribution imaging [126].

In addition, Bisht et al. (2016) conducted a study in which they examined the nanoencapsulation of EF24 within pegylated liposomes (Lipo-EF24) and evaluated its effectiveness in preclinical in vitro and in vivo pancreatic cancer models [14]. Their research revealed that encapsulated EF24 maintained a spherical morphology with an average diameter below 150 nm. In vitro, Lipo-EF24 treatment inhibited the growth of pancreatic cancer cells and induced apoptosis. When combined with the standard chemotherapy agent gemcitabine, Lipo-EF24 showed promise in inhibiting tumor growth in vivo [14]. These findings indicate that Lipo-EF24 holds potential as a solid foundation for future combinational therapies against pancreatic cancer, thanks to its encouraging therapeutic efficacy and low toxicity. Further, Sun et al. (2006) also synthesized EF24-tripeptide, a chloromethyl ketone, which shows potential for specific biomedical or therapeutic applications, particularly in the context of cancer therapy. However, more detailed research is needed to elucidate its full range of characteristics and applications [43].

## 5. Data and Notes from Preclinical Studies Pertaining to EF24

Based on the data from the conducted preclinical studies, EF24 offers several potential advantages when compared to other CUR analogues:EF24 has been designed to enhance its bioavailability, making it easier for the body to absorb and utilize.EF24 is designed to be more stable than CUR, ensuring a longer shelf life and improved effectiveness in various formulations.EF24 may exhibit a higher degree of specificity towards cancer cells, leading to reduced damage to healthy cells.EF24 has been reported to have superior anti-inflammatory effects in some studies.EF24 is known for its potent inhibition of the NF-κB signaling pathway, which is associated with inflammation, immunity, and cancer.EF24 may enhance the effectiveness of other cancer treatments when used in combination with other drugs.EF24 can be incorporated into drug delivery systems to further improve its delivery to specific tissues or cells.Researchers can modify the structure of EF24 to target specific conditions or diseases, making it a versatile compound for drug development.EF24 faces challenges due to its poor aqueous solubility and quick degradation in biological environments.

It is important to note that while EF24 shows several advantages over natural CUR, clinical research and long-term safety data are necessary to confirm its efficacy and safety for various medical applications. Furthermore, the choice between EF24 and other CUR analogues may depend on the specific needs of a particular application.

## 6. Future Directions

The exploration of EF24’s anticarcinogenic potency is an exciting area of research that holds immense promise for cancer therapy. Building on the existing knowledge, several future directions can further advance our understanding and clinical application of EF24 as a potent anticancer agent.

Conducting well-designed and rigorous clinical trials is essential to evaluating the safety and efficacy of EF24 in humans. Large-scale, multicenter trials across diverse cancer types should be undertaken to determine optimal dosages, treatment regimens, and potential combination therapies with conventional treatments. This will enable the transition of EF24 from preclinical to clinical settings.

Although the current understanding of EF24’s mechanisms is substantial, further research is needed to unravel its intricate molecular targets and signaling pathways. Identifying specific molecular targets and understanding the precise mechanisms of action will allow for more targeted therapeutic approaches and the development of personalized treatment strategies.

The emergence of drug resistance is a significant challenge in cancer treatment. Investigating the potential of EF24 to reverse or mitigate drug resistance, as well as its combination with other targeted therapies or immunotherapies, could hold the key to more effective and long-lasting anticancer interventions.

Continued research into nanoformulation approaches is crucial to optimizing EF24’s delivery to tumor sites while minimizing off-target effects. Fine-tuning the physicochemical properties of nanoformulations, enhancing their stability, and ensuring controlled drug release will improve EF24’s therapeutic index and maximize its potential.

The identification and validation of predictive biomarkers for EF24 response can aid in patient stratification, enable personalized treatment plans, and enhance treatment outcomes. Exploring novel imaging techniques or liquid biopsy-based approaches to monitor drug response and disease progression will contribute to more effective therapeutic management.

Investigating EF24’s synergistic potential with other existing therapies, such as immunotherapy or targeted agents, may lead to enhanced treatment outcomes and reduced side effects. Combinatorial approaches could offer a comprehensive and tailored treatment strategy for individual patients.

Utilizing advanced preclinical models, such as patient-derived xenografts and organoids, will provide more clinically relevant insights into EF24’s efficacy and safety profile. Such models can better predict responses in humans, expediting the drug development process.

Understanding the genetic variability that influences EF24 metabolism and response will aid in optimizing dosing and minimizing adverse effects. Comprehensive toxicity studies should be conducted to ensure EF24’s safety and establish appropriate dose escalation protocols.

The continued exploration of EF24’s anticarcinogenic potency through clinical trials, mechanistic studies, nanotechnology optimization, and combination therapies offers great promise for the future of cancer treatment. By addressing current limitations and pursuing innovative avenues of research, EF24 has the potential to become a valuable addition to the wide arsenal of anticancer therapeutics, ultimately improving patient outcomes and quality of life.

## 7. Conclusions

The data in published literature strongly support the antitumorigenic effect of EF24. The analyzed studies demonstrate that EF24 leads to a remarkable suppression of tumor growth in various experimental models. EF24 treatment resulted in a significant increase in apoptotic cell death within tumor tissues. This indicates that EF24 has the ability to induce programmed cell death, specifically in cancer cells. Numerous studies reported a significant decrease in cancer cell proliferation upon EF24 treatment. This suggests that EF24 effectively suppresses the abnormal and uncontrolled growth of cancer cells, which is a key characteristic of tumorigenesis. Recent reports also revealed that EF24 inhibits angiogenesis, as evidenced by reduced microvessel density within tumor tissues. This antiangiogenic effect contributes to the overall antitumorigenic activity of EF24. Molecular analysis demonstrated that EF24 regulates several critical signaling pathways associated with cancer development and progression. EF24 treatment led to the downregulation of oncogenic markers and the upregulation of tumor suppressor genes, indicating its ability to disrupt cancer-promoting pathways. Further investigation and clinical trials are warranted to fully explore the therapeutic utility of EF24 in combating tumorigenesis.

### Limitations

This study did not explore a wide range of doses or concentrations of EF24. Different effects, such as pro-oxidative effects or cytotoxicity, can occur at higher doses, which may limit its application or safety. This study did not introduce a comprehensive analysis of interactions with other drugs. Cellular and molecular interactions are intricate, and EF24’s effects can be influenced by various factors that may not all be accounted for in this study. This study may not extensively explore the potential of combining EF24 with other drugs, which can be a valuable strategy in cancer treatment. Future research is needed to further elucidate EF24’s therapeutic potential and to address its limitations for a more comprehensive understanding of its benefits and drawbacks.

## Figures and Tables

**Figure 1 cancers-15-05478-f001:**
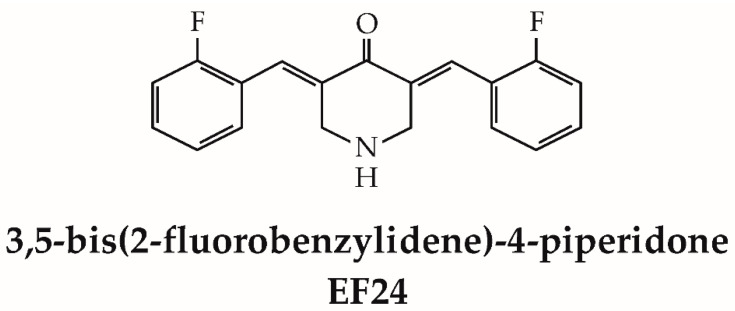
Structure of EF24.

**Figure 2 cancers-15-05478-f002:**
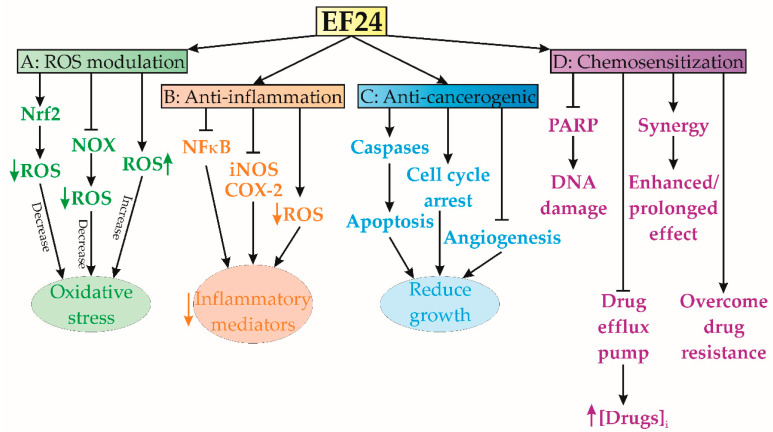
Summary of EF24 effects on cell viability and signaling. (**A**) Oxidative stress modulation; (**B**) anti-inflammatory effects; (**C**) anticancerogenic effects; (**D**) sensitization to other therapeutic agents. COX-2—cyclooxygenase-2, iNOS—inducible nitric oxide synthase; NFκB—nuclear factor kappa B, NOX—NADPH oxidase, Nrf2—nuclear factor erythroid 2-related factor 2, PARP—poly(ADP-ribose) polymerase, ROS—reactive oxygen species.

**Figure 3 cancers-15-05478-f003:**
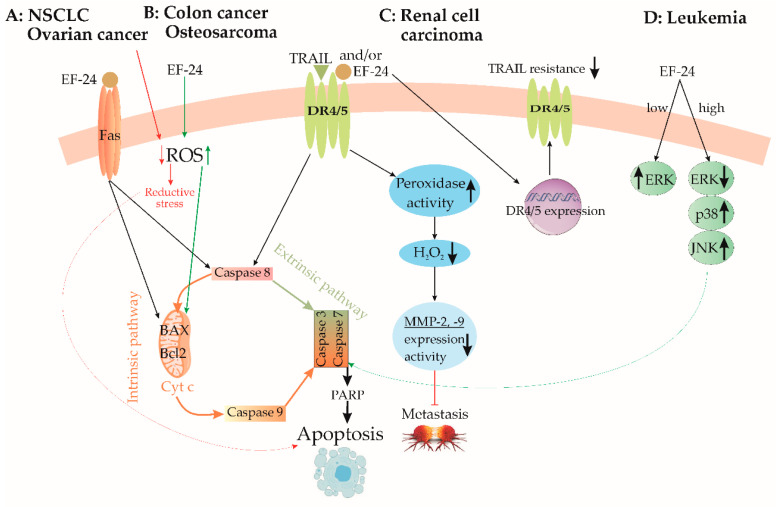
Pathways of activation of caspase signaling by EF24. (**A**) In non-small cell lung cancer and ovarian cancer, EF24 initiates apoptosis by the Fas receptor/intrinsic caspase pathway. In ovarian cancer, EF24 induces apoptosis further by reducing ROS production and reductive stress. (**B**) Oppositely, in colon cancer and osteosarcoma, EF24 increases ROS production, which activates the caspase pathway. (**C**) In renal cell carcinoma, EF24 induces apoptosis via the extrinsic caspase pathway, decreases metastasis by MMP-2 and -9 suppression, and improves drug sensitivity. (**D**) In leukemia cell lines, only a high concentration of EF24 induces apoptosis by the ERK/JNK mechanism. BAX—Bcl-2-like protein 4, Bcl-2—B-cell lymphoma 2, DR4/5—death receptors 4/5, ERK—extracellular signal-regulated kinase, JNK—c-Jun N-terminal kinase, MMP-2—matrix metalloproteinase-2, MMP-9—matrix metalloproteinase-9, PARP—poly(ADP-ribose) polymerase, ROS—reactive oxygen species, TRAIL—tumor necrosis factor-related apoptosis-inducing ligand, ↑—upregulation, ↓—downregulation.

**Figure 4 cancers-15-05478-f004:**
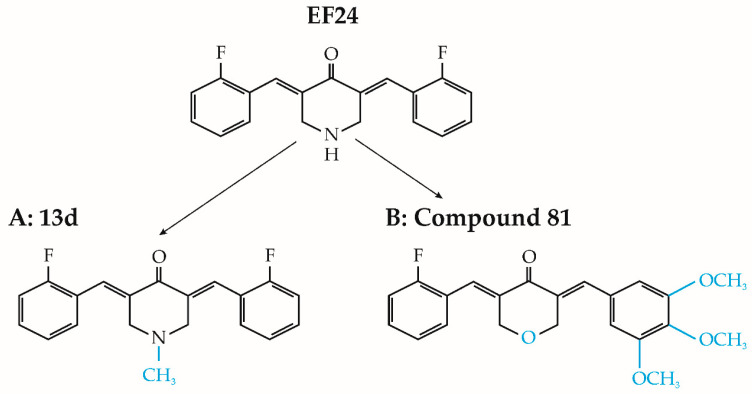
Chemical structure of EF24 analogs 13d and compound 81 (modifications are in blue).

**Figure 5 cancers-15-05478-f005:**
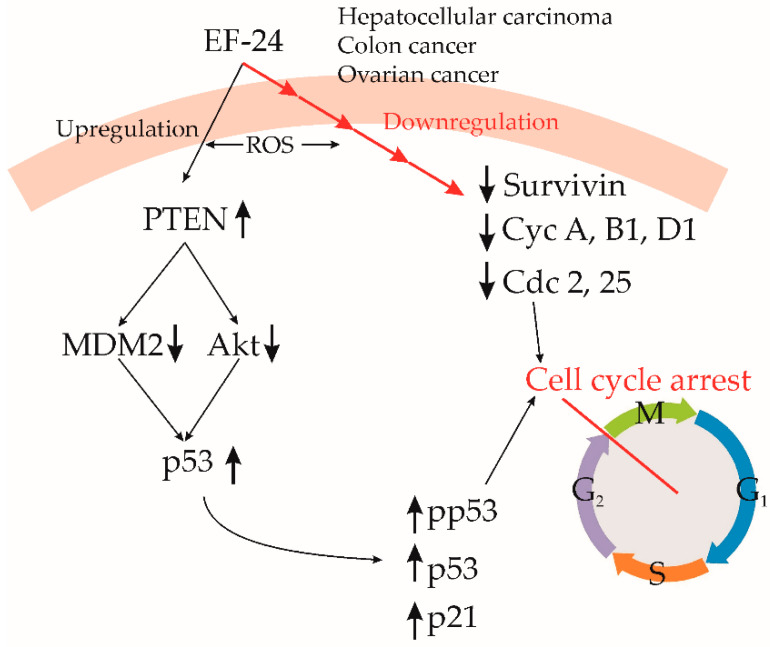
Cell cycle regulation by EF24. EF24 downregulates several proteins, which regulates cell cycle progression, resulting in cell cycle arrest. It decreases degradation of PTEN, which decreases MDM2 and Akt activity, preserving p53. In numerous cancer cell lines, EF24 causes cell cycle arrest at the G2/M stage. Cdc—cell division control, Cyc—cyclin, MDM2—mouse double minute 2, PTEN—phosphatase and tensin homolog, ROS—reactive oxygen species, ↑—upregulation, ↓—downregulation.

**Figure 6 cancers-15-05478-f006:**
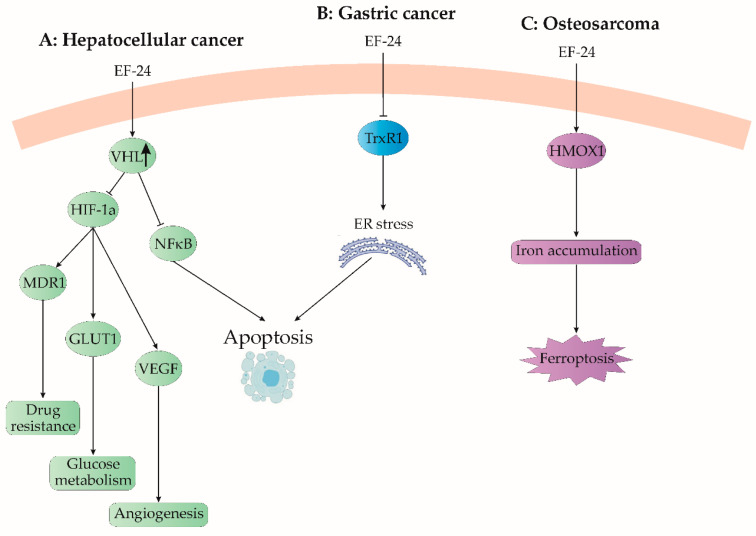
EF24 modulates some uncommon signaling pathways. (**A**) In hepatocellular cancer, EF24 upregulates von Hippel-Lindau (VHL) protein, a ubiquitin ligase, which decreases expression of hypoxia-induced genes, such as multidrug resistance 1 (MDR1), glucose transporter 1 (GLUT1), and vascular endothelial growth factor (VEGF). As a result, EF24 decreases drug resistance, modulates glucose metabolism, and reduces angiogenesis in growing cancer. (**B**) In gastric cancer, EF24 inhibits the thioredoxin system, resulting in overstorage of unfolded proteins in the endoplasmic reticulum (ER) and ER stress-induced apoptosis. (**C**) EF24 causes cell death in osteosarcoma by upregulation of heme oxygenase-1 (HMOX1). Accumulated iron induces ferroptosis in cancer cells. ↑—upregulation.

**Figure 7 cancers-15-05478-f007:**
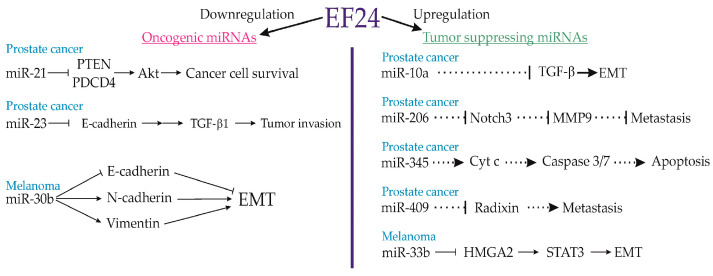
miRNAs involved in antitumor effects of EF24. EF24 downregulates some oncogenic miRNAs, which results in decreased cell viability and reduced loss of differentiation and conversion to tumor cells. At the same time, EF24 upregulates several tumour-suppressing miRNAs, which induces apoptosis, decreases the metastatic rate, and favour differentiation over tumorigenesis. EMT—epithelial–mesenchymal transition, HCG11—HLA complex group 11, HMGA2—high mobility group AT-hook 2, lncRNA—long non-coding RNAs, miR—microRNA, MMP-2—matrix metalloproteinase-2, MMP-9—matrix metalloproteinase-9, STAT3—signal transducer and activator of transcription 3, TGF-β1—transforming growth factor beta1.

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
