# Peer review of "Anticarcinogenic Potency of EF24: An Overview of Its Pharmacokinetics, Efficacy, Mechanism of Action, and Nanoformulation for Drug Delivery"

_cancers, 2023, doi:10.3390/cancers15225478_

Round 1

Reviewer 1 Report

Comments and Suggestions for Authors

Sazdova et al presented an overview on anticancer potential of curcumin derivative,  EF24. Authors covered the pharmacological properties emphasizing on anticancer potential apart from the well-known antiinflammatory activities. The manuscript also highlighted nanoformulation, mechanism of actions drug delivery and pharmacokinetic aspects.

The manuscript can be accepted after minor revision as suggested below:-

1. Since the poor absorption of the parent curcumin, various derivatives like EF-24 and dehydrozingerone were developed.  In the 1st para of introduction section, Authors should mention few available derivatives of curcumin and their drawbacks by discussion the rationale behind the selection of EF-24. Related references as mentioned below can  be cited:-

A. 10.1021/acsmedchemlett.6b00088 

B. 10.1021/np060252z

2. Authors can mention the toxicity profiling of EF-24 after the excretion section of pharmacokinetics

3. Section 4.2. drug delivery system needs to be elaborated with recent literature related to novel delivery systems.

4. Data and notes on clinical and preclinical studies pertaining to EF-2R4 to be mentioned in a separate paragraph before the future direction  

Author Response

  1. Since the poor absorption of the parent curcumin, various derivatives like EF-24 and dehydrozingerone were developed. In the 1st para of introduction section, Authors should mention few available derivatives of curcumin and their drawbacks by discussion the rationale behind the selection of EF-24. Related references as mentioned below can be cited:
  2. 10.1021/acsmedchemlett.6b00088
  3. 10.1021/np060252z

Answer: Page 2 (Section 1.1. Structurally-related mechanisms of action, text in red, lines 84-94, included 6 new references).

  1. Authors can mention the toxicity profiling of EF-24 after the excretion section of pharmacokinetics

Answer: Pages 8 and 9 (Section 2.5. Cytotoxicity, text in red, lines 382-399, included 2 new references).

  1. Section 4.2. Drug delivery system needs to be elaborated with recent literature related to novel delivery systems.

Answer: Page 24 (Section 4.2. EF24 Drug Delivery Systems, text in red, lines 1105-1139).

  1. Data and notes on clinical and preclinical studies pertaining to EF-24 to be mentioned in a separate paragraph before the future direction.

Answer: Page 25 (Section 5. Data and notes from preclinical studies pertaining to EF-24, text in red, lines 1152-1179).

Reviewer 2 Report

Comments and Suggestions for Authors

The review mainly discusses the anticarcinogenic activity of EF24. Additionally, the authors reviewed the pharmacokinetics, efficacy, mechanism of action, and effective drug delivery. This is a comprehensive study; however, revisions are required before publishing.

1.   Abstract – include values.

2.   Introduction – Are there any previous records on the toxicity of EF24?

3.   The objective of the study is not very clear. The authors should emphasize that.

4.   Being a lipophilic substance, what would be the side effects of using this as a drug? Emphasis on this.

5.   Is there cross-talking between ROS and other pathways?

6.   Anti-inflammatory activity – Highlight whether this compound is mainly pro- or anti-inflammatory.

7.   Anticancer- Emphasis on genetic pathway of action. Also, highlight whether been pro- or pre-apoptotic.

8.   Why are the authors focusing on specific cancers?

9.   4.2- drug delivery system: More information should be given. Also, what were the effective nano-synthesis materials used?

10.  Include limitations.

Author Response

The review mainly discusses the anticarcinogenic activity of EF24. Additionally, the authors reviewed the pharmacokinetics, efficacy, mechanism of action, and effective drug delivery. This is a comprehensive study; however, revisions are required before publishing.

  1. Abstract – include values.

Answer: Page 1 and 2 (Section Abstract, text in red, lines 46-50).

  1. Introduction – Are there any previous records on the toxicity of EF24?

Answer: Page 9 (Section 2.5. Cytotoxicity, text in red, lines 382-399, included 2 new references).

  1. The objective of the study is not very clear. The authors should emphasize that.

Answer: Page 9 (Separate Paragraph after Section 2.5., text in red, lines 401-405).

  1. Being a lipophilic substance, what would be the side effects of using this as a drug? Emphasis on this.

Answer: Page 3 (Section 1.2. Lipophilic properties, text in red, lines 120-122).

  1. Is there cross-talking between ROS and other pathways?

Answer: Pages 4 and 5 (Section 1.3. EF24-mediated ROS modulation, text in red, lines 169-175 and 189-202, certain cross-talk mechanisms are described in different sections of the study, i.e., Section 1.4. Anti-inflammatory effects; 1.5. Anticarcinogenic activity, etc.).

The cross-talk between ROS and various signaling pathways induced by EF-24 underscores its complex mechanism of action. While the generation of ROS may initially appear as a potential drawback due to oxidative stress, EF-24 utilizes this phenomenon to its advantage by affecting multiple pathways, ultimately promoting anti-cancer and anti-inflammatory effects. Nevertheless, the specific effects and consequences of this cross-talk may vary depending on the cellular context and the concentrations of EF-24 used, and further research is needed to understand these interactions fully.

  1. Anti-inflammatory activity – Highlight whether this compound is mainly pro- or anti-inflammatory.

Answer: Page 5 (Section 1.4. Anti-inflammatory effects, text in red, lines 176-188).

It is important to note that while EF-24 primarily exhibits anti-inflammatory properties, its effects can vary depending on the specific context and concentration used. In certain situations, such as high concentrations or specific cellular conditions, EF-24 may have pro-oxidative effects, which can indirectly contribute to inflammation. Therefore, understanding the precise conditions and dosage of EF-24 is crucial to harnessing its anti-inflammatory potential effectively.

  1. Anticancer- Emphasis on genetic pathway of action. Also, highlight whether been pro- or pre-apoptotic.

Answer: EF-24 was shown to exhibits anti-cancer properties through several genetic pathways of action. Additionally, it has been associated with promoting apoptosis, the programmed cell death of cancer cells, making it a pro-apoptotic agent in many cases. EF-24 is known to inhibit the nuclear factor-kappa B (NF-κB) pathway. This pathway is a central regulator of inflammation and cell survival. Inhibition of NF-κB can lead to the downregulation of pro-inflammatory genes and the suppression of cell survival mechanisms.

The comprehensive analysis of genetic pathways and their associated apoptotic features specific to each type of cancer is presented in Section 3, entitled "Antitumorigenic Effects and Mechanisms." This section encompasses all subsections from 3.1 to 3.16.

  1. Why are the authors focusing on specific cancers?

Answer: The effects of EF-24 can vary depending on the specific cellular or tissue context. Therefore, getting images of published data for all specific cancers addressed in the study may help in better understanding of EF-24’s anticancerogenic effects.

  1. 4.2- Drug delivery system: More information should be given. Also, what were the effective nano-synthesis materials used?

Answer: Page 24 (Section 4.2. EF24 Drug Delivery Systems, text in red, lines 1105-1139).

  1. Include limitations.

Answer: Page 27 (Section Limitations, text in red, lines 1244-1253).

Reviewer 3 Report

Comments and Suggestions for Authors

The paper is well-written and brings important knowledge to the field. After some corrections, I would recommend publication.

A section discussing the potential toxicities of this analog is pivotal to the general context of using EF24 in clinics. Please include

Drug delivery section is not well explored. Due to the low solubility of curcumin and its derivatives, the use of drug delivery systems is very promising and relevant. The author must explain better the advantages of nanoparticles In this particular case and discuss more papers on this matter.  

Author Response

The paper is well-written and brings important knowledge to the field. After some corrections, I would recommend publication.

  1. A section discussing the potential toxicities of this analog is pivotal to the general context of using EF24 in clinics.

Answer: Page 9 (Section 2.5. Cytotoxicity, text in red, lines 382-399, included 2 new references).

  1. Drug delivery section is not well explored. Due to the low solubility of curcumin and its derivatives, the use of drug delivery systems is very promising and relevant. The author must explain better the advantages of nanoparticles In this particular case and discuss more papers on this matter.

Answer: Page 24 (Section 4.2. EF24 Drug Delivery Systems, text in red, lines 1105-1139).

We would like to highlight that while there is a wealth of published research on drug delivery systems for curcumin, there is a limited number of studies dedicated to EF-24 delivery systems. This indicates a potential for further enhancement of its delivery, which remains open for investigation. We hope that further research in this area will lead to new improvements in its effectiveness against various types of cancer.

Reviewer 4 Report

Comments and Suggestions for Authors

The subject is very interesting and has been reviewed a few times recently, some errors should be corrected regarding the citations in the text and other details in the list of bibliographic references.

Comments on the Quality of English Language

Minor editing of the English language required

Author Response

  1. The subject is very interesting and has been reviewed a few times recently, some errors should be corrected regarding the citations in the text and other details in the list of bibliographic references.

Answer: We agree with Reviewer 4 and accepted all indicated errors.

  1. Minor editing of the English language required

Answer: Minor editing of the language was also done.

Round 2

Reviewer 3 Report

Comments and Suggestions for Authors

The authors tried to improve the paper according to my comments. However, the first question was not fully resolved or understood by the authors. Section 2.5 discusses cytotoxicity in tumor cells but another important issue is the toxicity in normal cells and systemic toxicity after injection by biochemical, hematological, and histopathological analyses. Please include a discussion about this aspect.

Author Response

Second response to Reviewer No 3

Dear Reviewer, I express my sincere gratitude for your valuable suggestion regarding including facts related to EF-24 toxicity in normal cells and systemic toxicity, which will undoubtedly enhance the manuscript's overall quality. I appreciate your diligence. Please refer to paragraph 2.5, "Toxicity of EF-24," on page 9, lines 401–417, for the detailed discussion on EF-24 toxicity concerning normal non-cancerogenic tissues (one new reference number 47 was also introduced in the text).

Special thanks to the esteemed editor for your dedicated efforts in refining our manuscript to a higher quality standard. We believe that this time, our submission meets all necessary criteria for acceptance in Cancer.

Feel free to reach out should you have any queries. Regarding the figures, I'd like to clarify that no “copyright permissions” are required as they were originally created specifically for this manuscript's subject matter. Thank you once more for your attention and спент time.

Best regards,

Prof. Avtanski Dimiter
